# Learning (Very) Simple Generative Models Is Hard

**Sitan Chen**
UC Berkeley
sitanc@berkeley.edu

**Jerry Li**
Microsoft Research
jerrl@microsoft.com

**Yuanzhi Li**
CMU
yuanzhil@andrew.cmu.edu

## Abstract

Motivated by the recent empirical successes of deep generative models, we study the computational complexity of the following unsupervised learning problem. For an unknown neural network $F : \mathbb{R}^d \to \mathbb{R}^{d'}$, let $D$ be the distribution over $\mathbb{R}^{d'}$ given by pushing the standard Gaussian $\mathcal{N}(0, \mathrm{Id}_d)$ through $F$. Given i.i.d. samples from $D$, the goal is to output *any* distribution close to $D$ in statistical distance.

We show under the statistical query (SQ) model that no polynomial-time algorithm can solve this problem even when the output coordinates of $F$ are one-hidden-layer ReLU networks with $\log(d)$ neurons. Previously, the best lower bounds for this problem simply followed from lower bounds for *supervised learning* and required at least two hidden layers and $\mathrm{poly}(d)$ neurons [CGKM22, DV21].

The key ingredient in our proof is an ODE-based construction of a compactly supported, piecewise-linear function $f$ with polynomially-bounded slopes such that the pushforward of $\mathcal{N}(0, 1)$ under $f$ matches all low-degree moments of $\mathcal{N}(0, 1)$.

## 1 Introduction

In recent years, deep generative models such as variational autoencoders, generative adversarial networks, and normalizing flows [GPAM+14, KW13, RM15] have seen incredible success in modeling real world data. These work by learning a parametric transformation (e.g. a neural network) of a simple distribution, usually a standard normal random variable, into a complex and high-dimensional one. The learned distributions have been shown to be shockingly effective at modeling real world data. The success of these generative models begs the following question: when is it possible to learn such a distribution? Not only is this a very natural question from a learning-theoretic perspective, but understanding this may also lead to more direct methods to learn generative models for real data.

More formally, we consider the following problem. Let $D$ be the unknown pushforward distribution over $\mathbb{R}^d$ given by $f(g)$, where $g \sim \mathcal{N}(0, \mathrm{Id})$ is a standard normal Gaussian, and $f$ is an unknown feedforward neural network with non-linear (typically ReLU) activations. Such distributions naturally arise as the output of many common deep generative models in practice. The learner is given $n$ samples from $D$, and their goal is to output the description of some distribution which is close to $D$.

When $f$ is a one-layer network (i.e. of the form $f(g) = \mathrm{ReLU}(Wg)$), there are efficient algorithms for learning the distribution [WDS19, LLDD20]. However, this setting is unsatisfactory in many ways, as one-layer networks lack much of the complex structure that makes these generative models so appealing in practice. Indeed, when the neural network only has a single layer, the resulting distribution is similar to a truncated Gaussian, and one can leverage techniques developed for learning from truncated samples. Notably, this structure disappears even with two-layer neural networks. Even in the two-layer case, despite significant interest, very little is known about how to learn $D$ efficiently.

In fact, a recent line of work suggests that learning neural network pushforwards of Gaussians may be an inherently difficult computational task. Recent results of [DV21, CGKM22] show hardness of *supervised* learning from labeled Gaussian examples under cryptographic asssumptions, and the latter also demonstrates hardness for all statistical query (SQ) algorithms (see Section 1.3 for a more

detailed description of related work). These naturally imply hardness in the unsupervised setting (see supplement). However, these lower bound constructions still have their downsides. For one, all of these constructions require at least three layers (i.e. two hidden layers), and so leave open the possibility that efficient learning is possible when the neural network only has one hidden layer. Additionally, the resulting neural networks in these constructions are quite complicated. In particular, the size of the neural networks in these hard instances, that is, the number of hidden nonlinear activations for any output coordinate, must be polynomially large. This begs the natural question:

*Can we learn pushforwards of Gaussians under one-hidden-layer neural networks of small size?*

## 1.1 Our Results

We demonstrate strong evidence that despite the simplicity of the setting, this learning task is already computationally intractable. We show there is no polynomial-time statistical query (SQ) algorithm which can learn the distribution of $f(g)$, when $g \sim \mathcal{N}(0, \mathrm{Id})$ and each output coordinate of $f$ is a one-hidden-layer neural network of *logarithmic hidden size*. We formally define the SQ model in Definition 2; we note that it is well-known to capture almost all popular learning algorithms [FGR+17].

**Theorem 1.1** (informal, see Theorem 3.1). *For any $d > 0$, and any $C \geq 1$, there exists a family of one-hidden-layer neural networks $\mathcal{F}$ from $\mathbb{R}^d$ to $\mathbb{R}^{d^C}$ so that the following properties hold. For any $f \in \mathcal{F}$, let $D_f$ denote the distribution of $f(g)$, for $g \sim \mathcal{N}(0, \mathrm{Id})$. Then, we have that:*

- *For all $f \in \mathcal{F}$, $\mathrm{d}_{\mathrm{TV}}(\mathcal{N}(0, \mathrm{Id}), D_f) = \Omega(1)$,[1]*

- *Every output coordinate of $f$ is a sum of $O(\log d / \log \log d)$ ReLUs, with $\mathrm{poly}(d)$-bounded weights.*

- *Any SQ algorithm which can distinguish between $D_f$ and $\mathcal{N}(0, \mathrm{Id})$ with high probability for all $f \in \mathcal{F}$ requires $d^{\omega(1)}$ time and/or samples.*

In other words, there is a family of one-hidden-layer ReLU networks of logarithmic size whose corresponding pushforwards are statistically very far from Gaussian, yet no efficient SQ algorithm can distinguish them from a Gaussian. Note this implies hardness even of *improperly* learning such pushforwards: not only is it hard to recover the parameters of the network or output a network close to the underlying distribution $D_f$, but it is hard to learn *any* distribution close to $D_f$. In contrast, if one ignores issues of computational efficiency, one can easily learn networks in this family with polynomial sample complexity via hypothesis selection, as there are only $\mathrm{poly}(d)$ many parameters specifying any network in this family.

Since such networks are arguably some of the simplest neural networks with more than one layer, this suggests that learning even the most basic deep generative models may already be a very difficult task, at least without additional assumptions. Still, this is by no means the last word in this direction. Given the real world success of deep generative models, a natural and important direction is to identify natural conditions under which we can efficiently learn. We view our results as a first step towards understanding the computational landscape of this important learning problem, and our result *provides evidence that (strong) assumptions need to be made on $f$ for the pushforward $f(g)$ to be efficiently learnable, even in very simple two-layer cases.*

## 1.2 Our Techniques

Like many recent SQ lower bounds, ours follows the general framework which was introduced in [DKS17] and builds on [FGR+17]. Here one considers the following "non-Gaussian component analysis" task. We will consider a family of distributions, parametrized by a "hidden" direction $v \in \mathbb{S}^{d-1}$, which are Gaussian in every direction orthogonal to the hidden direction but which are very non-Gaussian along the hidden direction.

To formalize this, let $D$ be a distribution $D$ over $\mathbb{R}$ which is known to the learner and far in statistical distance from Gaussian. Given a unit vector $v$ in $d$ dimensions, let $P_v^D$ denote the distribution over $\mathbb{R}^d$ whose projection along $v$ is given by $D$ and whose projection in all directions orthogonal to $v$ is standard Gaussian. Given samples from some unknown distribution over $\mathbb{R}^d$, the goal is to

---

[1]$\mathrm{d}_{\mathrm{TV}}$ denotes total variation distance. A lower bound for Wasserstein distance also holds, see supplement.

decide whether the unknown distribution is $\mathcal{N}(0, \mathrm{Id}_d)$ or $P_v^D$ for some $v$. [DKS17] showed that if $D$'s moments match those of $\mathcal{N}(0, 1)$ up to some degree $m$, then under mild conditions, any SQ algorithm for this task requires at least $d^{\Omega(m)}$ queries (Lemma 3.3).

Suppose one could exhibit a one-hidden-layer ReLU network $f : \mathbb{R}^\ell \to \mathbb{R}$ such that the pushforward $D = f(\mathcal{N}(0, \mathrm{Id}))$ satisfied such properties. Then we can realize $P_v^D$ as a pushforward as follows. Let $U$ be a rotation mapping the first standard basis vector in $\mathbb{R}^d$ to $v$. Then consider the function $F : \mathbb{R}^{\ell+d-1} \to \mathbb{R}^d$ mapping $z$ to $U \cdot (f(z_1, \ldots, z_\ell), z_{\ell+1}, \ldots, z_{\ell+d-1})$. One can check that every output coordinate of $F$ is computed by a one-hidden-layer ReLU network with size essentially equal to that of $f$. By [DKS17], we would immediately get the desired SQ lower bound.

The main challenge is thus to construct such a network whose pushforward matches the low-degree moments of $\mathcal{N}(0, 1)$. It is not hard to ensure the existence of such $f$ with essentially *infinite* weights (Corollary 4.2 and Lemma 4.4). It is much less clear whether this is possible with *polynomially bounded* weights, and this is our primary technical contribution. We design and analyze a certain ODE which defines a one-parameter family of perturbations to $f$, such that the low-degree moments of the corresponding pushforwards remain unchanged over time. By evolving along this family over an inverse-polynomial time scale, we obtain a network with polynomially bounded weights whose pushforward matches the low-degree moments of $\mathcal{N}(0, 1)$. We defer the details to Section 4.

## 1.3 Related Work

A full literature review on the theory of learning deep generative models is beyond the scope of this paper (see e.g. the survey of [GSW$^+$21]). For conciseness we cover only the most relevant papers.

**Upper bounds.** In terms of upper bounds, much of the literature has focused on a different setting, where the goal is to understand when first order dynamics can learn toy generative models [FFGT17, DISZ17, GHP$^+$19, LLDD20, AZL21, JMGL22], which are much simpler than the ones we consider here. For learning pushforwards of Gaussians under neural networks with ReLU activations, algorithms with provable guarantees are only known when the network has no hidden layers [WDS19, LLDD20]. This is in contrast to the supervised setting, where fixed parameter tractable algorithms are known for learning ReLU networks of arbitrary depth [CKM22].

A different line of work seeks to find efficient learning algorithms when the activations are given by low degree polynomials [FFGT17, LD20, CLLZ22]. Arguably the closest to our work is [CLLZ22], which gives polynomial-time algorithms for learning low-degree polynomial transformations of Gaussians, in a smoothed setting. It is a very interesting open question if similar smoothed assumptions can be leveraged to circumvent our lower bound when the activations are ReLU. Unfortunately, these papers heavily leverage the nice moment structure of low-degree Gaussian polynomials, and it is unclear how their techniques can generalize to different activations.

**Lower bounds.** Much of the literature on lower bounds for learning neural networks has focused on the supervised setting, where a learner is given labeled examples $(x, f(x))$, and the goal is to output a good predictor. There are many lower bounds known in the distribution-free setting [BR92, Vu98, KS09, LSSS14, DV20], however, these do not transfer over to our (unsupervised) setting. When $x$ is Gaussian, the aforementioned work of [CGKM22] derives hardness for learning two-hidden-layer networks with polynomial size for all SQ algorithms, as well as under cryptographic assumptions (see also [DV21]). It is not hard to show (see supplement) that this lower bound immediately implies a lower bound for the unsupervised problem. In the supervised setting, lower bounds are also known against restricted families of SQ [VW19, GGJ$^+$20, DKKZ20, SVWX17], when there are adversarially noisy labels [KK14, DKZ20, GGK20, SZB21], and in discrete settings [Val84, Kha95, AK95, Fel09, CGV15, DGKP20, AAK21], but to our knowledge, these results do not transfer to our setting.

The literature on lower bounds for the unsupervised problem we consider here is much sparser. Besides [DV21, CGKM22], we also mention the recent work of [CLLM22] that studies whether achieving small Wasserstein GAN loss implies distribution learning. A corollary of their results is cryptographic hardness for learning pushforwards of Gaussians under networks with constant depth and polynomial size, but only when the learner is given by a Lipschitz ReLU network discriminator. However, this does not rule out efficient algorithms which do not output such Lipschitz discriminators.

Finally, we remark that the family of hard distributions we construct can be thought of as a close cousin of the "parallel pancakes" construction of [DKS17]. This and slight modifications thereof are mixtures of Gaussians which are known to be computationally hard both in the SQ model [DKS17, BLPR19] and under cryptographic assumptions [BRST21, GVV22].

**SQ lower bounds via ODEs.** We remark that in a very different context, [DKZ20] also used an ODE to design a moment-matching construction. While our approach draws inspiration from theirs, an important difference is that they use their ODE as a "size reduction" trick to construct a step function $f : \mathbb{R} \to \{\pm 1\}$ with a small number of linear pieces such that $\mathbb{E}_{g \sim \mathcal{N}(0,1)}[f(g)g^k] = 0$ for all small $k$, while we use our ODE as a "weight reduction" trick to construct a continuous neural network $f : \mathbb{R} \to \mathbb{R}$ with bounded weights such that $\mathbb{E}_{g \sim \mathcal{N}(0,1)}[f(g)^k] = \mathbb{E}_{g \sim \mathcal{N}(0,1)}[g^k]$. The form of the moments $\mathbb{E}_{g \sim \mathcal{N}(0,1)}[f(g)g^k]$ they consider is simpler than in our setting, and while they essentially run their ODE to singularity and use non-quantitative facts like the invertibility of a certain Jacobian, we only run our ODE for a finite horizon and need to carefully control the condition number of the Jacobian arising in our setting over this horizon (e.g. Lemma 4.8).

## 2 Technical Preliminaries

We freely abuse notation and use the same symbols to denote probability distributions, their laws, and their density functions. Given a distribution $A$ over a domain $\Omega$ and a function $f : \Omega \to \Omega'$, we let $f(A)$ denote the *pushforward* of $A$ through $f$, that is, the distribution $A'$ of the random variable $f(z)$ for $z \sim A$. Given distributions $p, q$ for which $p$ is absolutely continuous with respect to $q$, let $\chi^2(p, q)$ denote their chi-squared divergence. Let $p \star q$ denote the convolution of $p$ and $q$. Also, we use $\|\cdot\|_p$ to denote $\ell^p$ norm, omitting the subscript when $p = 2$. $\sigma_{\min}(\cdot)$ denotes minimum singular value. Given an invertible matrix $M$, we use $M^{-\top}$ to denote the matrix $(M^\top)^{-1} = (M^{-1})^\top$.

Henceforth $\mathbb{E}_g[\cdot]$ will always denote $\mathbb{E}_{g \sim \mathcal{N}(0,\mathrm{Id})}[\cdot]$, where the dimensionality is implicit from context. Let $\gamma_{\sigma^2}(x) \triangleq \frac{1}{\sigma\sqrt{2\pi}} e^{-x^2/(2\sigma^2)}$. Given $S \subset \mathbb{R}$, we use $\gamma_{\sigma^2}(S)$ to denote $\int_{-\infty}^\infty \gamma(x) \cdot \mathbb{1}[x \in S]\, \mathrm{d}x$. When $\sigma = 1$, we omit the subscript $\sigma^2$. We will also use $\gamma^{(d)}(x)$ to denote the density of $\mathcal{N}(0, \mathrm{Id}_d)$.

**Definition 1** (One-hidden-layer ReLU networks). *We say that $g : \mathbb{R}^d \to \mathbb{R}$ is a one-hidden-layer ReLU network with size $S$ and $W$-bounded weights if there exist $w_1, \ldots, w_S \in \mathbb{R}^d$, $b_1, \ldots, b_S \in \mathbb{R}$, and $s_1, \ldots, s_S \in \{\pm 1\}$ for which $g(x) = \sum_{i=1}^S s_i \mathrm{ReLU}(\langle w_i, x \rangle + b_i)$ and $\|w_i\|, |b_i| \le W$ for all $i \in S$.[2] Here $\mathrm{ReLU}(z) \triangleq \max(0, z)$. Given $f : \mathbb{R}^d \to \mathbb{R}^{d'}$, together with a distribution $A$ over $\mathbb{R}^d$, we say that $f(A)$ is a one-hidden-layer ReLU network pushforward of $A$ with size $S$ and $W$-bounded weights if each coordinate of $f$ is of this form.*

**Fact 2.1** ([GMSR20]). *If $V \in \mathbb{R}^{n \times n}$ is a Vandermonde matrix with nodes $z_1, \ldots, z_n$, that is, $V_{i,j} = z_j^{i-1}$, and $\{z_i\}$ are $\zeta$-separated, then $\sigma_{\min}(V) \ge \frac{1}{n} \cdot \Omega(\zeta)^{n-1}$.*

**Theorem 2.2** (Peano's existence theorem, see e.g. Theorem 2.1 from [Har02]). *For $T, r > 0$ and $y_0 \in \mathbb{R}^n$, let $B \subset \mathbb{R} \times \mathbb{R}^n$ be the parallelepiped consisting of $(t, y)$ for which $0 \le t \le T$ and $\|y - y_0\|_\infty \le r$. If $f : B \to \mathbb{R}$ is continuous and satisfies $|f(t, y)| \le M$ for all $(t, y) \in B$, then the initial value problem $\{y'(t) = f(t, y)$ and $y(0) = y_0\}$ has a solution over $t \in [0, \min(T, r/M)]$.*

**Definition 2** (Statistical queries). *Given distribution $D$ over $\mathbb{R}^d$ and parameters $\tau, t > 0$, a $\mathrm{STAT}(\tau)$ oracle takes in any query of the form $f : \mathbb{R}^d \to [-1, 1]$ and outputs a value from $[\mathbb{E}_{x \sim D}[f(x)] - \tau, \mathbb{E}_{x \sim D}[f(x)] + \tau]$, while a $\mathrm{VSTAT}(t)$ oracle takes in any query of the form $f : \mathbb{R}^d \to [0, 1]$ and outputs a value from $[\mathbb{E}_{x \sim D}[f(x)] - \tau, \mathbb{E}_{x \sim D}[f(x)] + \tau]$ for $\tau = \max(1/t, \sqrt{\mathbb{V}_{x \sim D}[f(x)]/t})$.*

## 3 Statistical Query Lower Bound

Our main result is the following lower bound in this model for learning ReLU network pushforwards:

**Theorem 3.1.** *Let $d \in \mathbb{N}$ be sufficiently large. Any SQ algorithm which, given SQ access to an arbitrary one-hidden-layer ReLU network pushforward of $\mathcal{N}(0, \mathrm{Id}_d)$ of size $O(\log d / \log \log d)$ with $\mathrm{poly}(d)$-bounded weights, outputs a distribution which is $O(1)$-close in $\mathrm{d}_{\mathrm{TV}}(\cdot)$ must make at least $d^{\Omega(\log d / \log \log d)}$ queries to either $\mathrm{STAT}(\tau)$ or $\mathrm{VSTAT}(1/\tau^2)$ for $\tau = d^{-\Omega(\log d / \log \log d)}$.*

---

[2] In practice, one can think of $S$ as the number of channels, e.g. in a deconvolution network.

Our proof will invoke the following key technical result whose proof we defer to Section 4. It exhibits a one-hidden-layer ReLU network $f : \mathbb{R}^2 \to \mathbb{R}$ with bounded weights under which the pushforward of $\mathcal{N}(0, 1)$ matches the low-degree moments of $\mathcal{N}(0, 1)$ to arbitrary precision, in addition to some other technical conditions that we need to formally establish our statistical query lower bound:

**Theorem 3.2.** *Fix any odd $m$ and $\nu, \sigma < 1$. There is a one-hidden-layer ReLU network $f^* : \mathbb{R}^2 \to \mathbb{R}$ of size $O(m)$ with weights at most $m^{O(m)}$ for which the pushforward $D \triangleq f^*(\mathcal{N}(0, \mathrm{Id}))$ satisfies*

*1. $|\mathbb{E}_{x \sim D}[x^k] - \mathbb{E}_{g \sim \mathcal{N}(0,1)}[g^k]| < \nu$ for all $k = 1, \ldots, m$*

*2. $\chi^2(D, \mathcal{N}(0, 1)) \leq \exp(O(m))/\sigma$*

*3. $\mathrm{d}_{\mathrm{TV}}(P_v^D, P_{v'}^D) \geq 1 - 2\sigma \log(1/\sigma) - m^{-\Omega(m)}$ for any $v, v' \in \mathbb{S}^{d-1}$ satisfying $|\langle v, v' \rangle| \geq 1/2$.*

The rest of the proof of our lower bound will then follow the framework introduced in [DKS17] and subsequently generalized in [DK20]. First, given a distribution $D$ over $\mathbb{R}$ and $v \in \mathbb{S}^{d-1}$, let $P_v^D$ denote the distribution over $\mathbb{R}^d$ with density $P_v^D(x) = D(\langle v, x \rangle) \cdot \gamma^{(d-1)}(x - \langle v, x \rangle v)$, that is, the distribution which is given by $D$ in the direction $v$ and is given by $\mathcal{N}(0, \mathrm{Id} - vv^\top)$ orthogonal to $v$. We use the following generic statistical query lower bound about learning such distributions, a proof of which we include in the supplement for completeness.

**Lemma 3.3.** *Let $m \in \mathbb{N}$ and $0 < C < 1/2$. Let $D$ be a distribution such that 1) $\chi^2(D, \mathcal{N}(0, 1))$ is finite, and 2) $|\mathbb{E}_{x \sim D}[x^k] - \mathbb{E}_{g \sim \mathcal{N}(0,1)}[g^k]| \leq \Omega(d)^{-(m+1)(1/4 - C/2)} \sqrt{\chi^2(D, \mathcal{N}(0, 1))}$ for all $k = 1, \ldots, m$.*

*Consider the set of distributions $\{P_v^D\}_{v \in \mathbb{S}^{d-1}}$ for $d \geq m^{\Omega(1/C)}$. If there is some $\epsilon > 0$ for which $\mathrm{d}_{\mathrm{TV}}(P_v^D, P_{v'}^D) > 2\epsilon$ whenever $|\langle v, v' \rangle| \leq 1/2$, then any SQ algorithm which, given SQ access to $P_v^D$ for an unknown $v \in \mathbb{S}^{d-1}$, outputs a hypothesis $Q$ with $\mathrm{d}_{\mathrm{TV}}(Q, P_v^D) \leq \epsilon$ needs at least $d^{m+1}$ queries to $\mathrm{STAT}(\tau)$ or to $\mathrm{VSTAT}(1/\tau^2)$ for $\tau \triangleq O(d)^{-(m+1)(1/4 - C/2)} \cdot \sqrt{\chi^2(D, \mathcal{N}(0, 1))}$.*

*Proof of Theorem 3.1.* By Theorem 3.2 applied with sufficiently large odd $m$ and sufficiently small $\sigma$, there exists a distribution $D = f^*(\mathcal{N}(0, \mathrm{Id}_2))$ over $\mathbb{R}$ for $f^* : \mathbb{R}^2 \to \mathbb{R}$ of size $O(m)$ with $m^{O(m)}$-bounded weights satisfying the hypotheses of Lemma 3.3 for $\epsilon = 0.49$, and $\chi^2(D, \mathcal{N}(0, 1)) \leq \exp(O(m))$ (note that we can absorb the $1/\sigma$ factor in Theorem 3.2 into $\exp(O(m))$ because we can take $\sigma = \exp(-\Theta(m))$). As long as $m \leq d^{O(C)}$, we conclude that an SQ algorithm for learning any distribution from $\{P_v^D\}_{v \in \mathbb{S}^{d-1}}$ to total variation distance $1/4$ must make at least $d^{m+1}$ queries to $\mathrm{STAT}(\tau)$ or $\mathrm{VSTAT}(1/\tau^2)$ for $\tau \triangleq O(d)^{-(m+1)(1/4 - C/2)} \cdot \exp(O(m))$. By taking $m = \Theta(\log d / \log \log d)$, we ensure that $m^{O(m)} \leq \mathrm{poly}(d)$. We're done by taking $C$ in Lemma 3.3 to be $C = 1/4$.

The proof of the theorem is complete upon noting that any distribution $P_v^D$ can be implemented as a pushforward of $\mathcal{N}(0, \mathrm{Id}_{d+1})$ under a one-hidden-layer ReLU network $F_v : \mathbb{R}^{d+1} \to \mathbb{R}^d$ of size $O(\log d / \log \log d)$ with $\mathrm{poly}(d)$-bounded weights. Let $U \in O(d)$ be a rotation mapping the first standard basis vector in $\mathbb{R}^d$ to $v$. Then for $F_v(z_1, \ldots, z_{d+1}) \triangleq U(f^*(z_1, z_2), z_3, \ldots, z_{d+1})$ we have that $F_v(\mathcal{N}(0, \mathrm{Id}_{d+1})) = P_v^D$ as desired. Furthermore, note that every output coordinate of $F_v(z_1, \ldots, z_{d+1})$ is a one-hidden-layer ReLU network of the form $\alpha f^*(z_1, z_2) + \langle u, (z_3, \ldots, z_{d+1}) \rangle$ for some vector $(\alpha, u) \in \mathbb{R}^d$. Note that the size of this network is two plus that of $f^*$, and its weights are also upper bounded by $\mathrm{poly}(d)$, so $F_v$'s output coordinates are of size $O(\log d / \log \log d)$. $\square$

*Remark* 3.4. Theorem 1.1 was stated with output dimension polynomially bigger than input dimension, whereas in our construction, the output dimension ($d$) is less than the input dimension ($d + 1$). One can get the former by a padding argument (i.e. by duplicating output coordinates) to give a generator with arbitrarily large polynomial stretch and such that the $d^{\log d / \log \log d}$ lower bound still applies.

# 4 Moment-Matching Construction

In this section we prove Theorem 3.2, the main technical ingredient in the proof of Theorem 3.1.

## 4.1 Moment-Matching With Unbounded Weights

We begin by making the simple observation that for one-hidden-layer networks with *unbounded weights*, it is easy to construct networks such that the pushforward of $\mathcal{N}(0, 1)$ under these networks

matches the moments of $\mathcal{N}(0,1)$. The starting point for this is the following standard construction. Roughly speaking, it gives a set of well-separated points on the real line and a distribution over them which matches low-order moments with $\mathcal{N}(0,1)$.

**Lemma 4.1** (Lemma 4.3 from [DKS17]). *For any $m \in \mathbb{N}$, there exist weights $\lambda_1, \ldots, \lambda_m \geq 0$ and points $h_1, \ldots, h_m \in \mathbb{R}$ for which*

1. *(Moments match) $\sum_{i=1}^m \lambda_i h_i^k = \mathbb{E}_g[g^k]$ for all $k = 0, \ldots, 2m-1$ (note that the special case of $k = 0$ implies $\sum_{i=1}^m \lambda_i = 0$).*

2. *(Points symmetric about origin) $h_1 \leq \cdots \leq h_m$ and $h_i = -h_{m-i+1}$ for all $1 \leq i \leq m$.*

3. *(Weights symmetric) $\lambda_1 \leq \cdots \leq \lambda_{\lceil m/2 \rceil}$ and $\lambda_i = \lambda_{m-i+1}$.*

4. *(Points bounded and separated) $\Omega(1/\sqrt{m}) \leq |h_i| \leq O(\sqrt{m})$ for all $1 \leq i \leq m$ and $\{h_i\}$ are $\Omega(1/\sqrt{m})$-separated.*

5. *(Weights not too small) $\min_i \lambda_i \geq e^{-cm}$ for an absolute constant $c > 0$.*

6. *(Central point and weight) If $m$ is odd, then $h_{(m+1)/2} = 0$ and $\lambda_{(m+1)/2} = \Theta(1/\sqrt{m})$.*

This immediately implies that there exists a *discontinuous* piecewise linear function $f : \mathbb{R} \to \mathbb{R}$ for which the pushforward $f(\mathcal{N}(0,1))$ matches the low-degree moments of $\mathcal{N}(0,1)$. The reason is that we can take $f$ to be a step function which takes on values given by the positions of the points $h_1, \ldots, h_m$ in Lemma 4.1. We can then take the steps to have lengths such that the probability that a standard Gaussian input to $f$ falls under the step given by some $h_i$ is precisely $\lambda_i$ (see Figure 1).

**Corollary 4.2.** *For any $m \in \mathbb{N}$, there is a partition of $\mathbb{R}$ into disjoint intervals $I_1, \ldots, I_m$, along with a choice of scalars $h_1, \ldots, h_m$, such that the step function $f : \mathbb{R} \to \mathbb{R}$ given by $f(z) = \sum_{i=1}^m h_i \cdot \mathbb{1}[z \in I_i]$ satisfies $\mathbb{E}_{x \sim f(\mathcal{N}(0,1))}[x^k] = \mathbb{E}_{g \sim \mathcal{N}(0,1)}[g^k]$ for all $k = 0, \ldots, 2m-1$.*

*Proof.* Let $\{\lambda_i\}$, $\{h_i\}$ be as in Lemma 4.1. As $\sum_i \lambda_i = 1$, there are intervals $\mathbb{R} = I_1 \sqcup \cdots \sqcup I_m$ for which $\gamma(I_i) = \lambda_i$. As $\mathbb{E}_{x \sim f(\mathcal{N}(0,1))}[x^k] = \sum_i \lambda_i h_i^k$, the claim follows by Part 1 of Lemma 4.1. $\square$

By infinitesimally perturbing $f$ in Corollary 4.2, we can ensure $f(\mathcal{N}(0,1))$ still *approximately* matches the low-degree moments of $\mathcal{N}(0,1)$ to arbitrary precision and that the linear pieces of $f$ have finite slopes, though some slopes will be arbitrarily large. The new function can thus be represented as a one-hidden-layer ReLU network, but unfortunately its weights will be arbitrarily large. The key challenge in the sequel is to design a better perturbation so the resulting $f$ has *polynomially bounded* slopes yet is such that $f(\mathcal{N}(0,1))$ matches the low-degree moments of $\mathcal{N}(0,1)$.

## 4.2 Bump Construction

Before we describe our perturbation scheme, we slightly modify the construction in Corollary 4.2. In place of a step function, we will consider a certain sum of *bump functions*, illustrated in Figure 2.

**Definition 3** (Bump functions). *Given $w, \epsilon > 0$ and $h, c \in \mathbb{R}$, define $T_c^{w,h,\epsilon} : \mathbb{R} \to \mathbb{R}$ by*

$$T_c^{w,h,\epsilon}(z) = \begin{cases} \frac{h}{\epsilon}(z - c + \epsilon + w) & \text{if } z \in [c - \epsilon - w, c - w] \\ h & \text{if } z \in [c - w, c + w] \\ -\frac{h}{\epsilon}(z - c - \epsilon - w) & \text{if } z \in [c + w, c + \epsilon + w] \\ 0 & \text{otherwise.} \end{cases}$$

Because these are piecewise linear functions, we can implement them as one-hidden-layer ReLU networks:

**Fact 4.3.** *Given $w, \epsilon > 0$ and $h, c \in \mathbb{R}$, $T_c^{w,h,\epsilon}$ can be implemented as a one-hidden-layer ReLU network with size $4$ and $W$-bounded weights for $W \leq \frac{h}{\epsilon} \max(1, |c| + \epsilon + w)$.*

In the next lemma, we show that we can replace the construction in Corollary 4.2, given by a step function, with a sum of well-separated bumps with infinitesimally small $\epsilon$ parameter. This will be the instance on which we will apply our perturbation scheme.

**Lemma 4.4.** *For any odd $m$ and $\nu < 1$, there are centers $c_1 \leq \cdots \leq c_{m-1}$, widths $w_1, \ldots, w_{m-1} > 0$, heights $h_1 \leq \cdots \leq h_{m-1} \in \mathbb{R}$, and parameter $\bar{\epsilon} > 0$ for which the following holds. Define the function $f : \mathbb{R} \to \mathbb{R}$ (see Figure 1) by $f(z) \triangleq \sum_{i=1}^{m-1} T_{c_i}^{w_i, h_i, \epsilon}(z)$ for any $0 \leq \epsilon < \bar{\epsilon}$. Then $f$ satisfies*

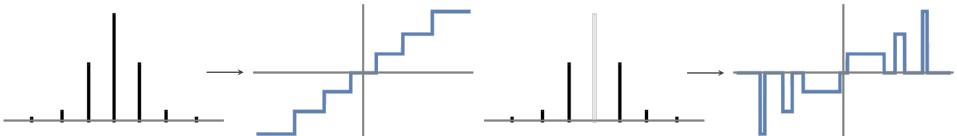

Figure 1: Left: construction from Lemma 4.1 gives rise to step function in Corollary 4.2. Right: removing central spike gives rise to sum of bumps in Lemma 4.4

1. *(Bumps are well-separated) For all $1 \le i < m - 1$, $c_i + m^{-3/2} \le c_{i+1}$.*

2. *(Moments match) $|\mathbb{E}_{x \sim f(\mathcal{N}(0,1))}[x^k] - \mathbb{E}_g[g^k]| < \nu$ for all $k = 1, \ldots, 2m - 1$.*

3. *(Symmetricity) $w_i = w_{m-i}$, $h_i = -h_{m-i}$, and $c_i = -c_{m-i}$ for all $1 \le i < m$.*

4. *(Bounded, separated heights) $\Omega(1/\sqrt{m}) \le |h_i| \le O(\sqrt{m})$ for all $1 \le i < m$, and $\{h_i\}$ are $\Omega(1/\sqrt{m})$-separated.*

5. *(Intervals not too thin) $\min_i \gamma([c_i - w_i, c_i + w_i]) \ge e^{-cm}$ for an absolute constant $c > 0$.*

6. *(Bounded endpoints) $|c_i| + w_i \le O(\log m)$ for all $1 \le i < m$.*

*Proof sketch, see supplement.* The idea is that when $m$ is odd, one of the points $h_i$ from Lemma 4.1 is zero and does not contribute to the moments. We want to design $f$ for which $f(\mathcal{N}(0,1))$ is the mixture of point masses given by removing this point and its corresponding weight. We take $f$ to have one linear piece for every one of the remaining point masses. By Part 6 of Lemma 4.1, these points have total weight $1 - \Theta(1/\sqrt{m})$, so by anticoncentration we can arrange these linear pieces to be $\sim m^{-3/2}$-separated, satisfying Part 1. The remaining parts follow easily from Lemma 4.1. □

Unfortunately the slopes in the function constructed in Lemma 4.4 are arbitrarily large if $\nu$ is arbitrarily small. The issue still remains of how to get a continuous piecewise-linear function whose slopes are *polynomially bounded* so that the corresponding ReLU network has polynomially bounded weights. As we illustrate next however, the $\Omega(m^{-3/2})$ spacing between the bumps in the definition of $f$ in Lemma 4.4 gives us sufficient "room" to carefully perturb $f$ to achieve this goal.

### 4.2.1  Some Estimates for Bump Moments

For convenience, define $M_{c,k}^{w,h,\epsilon} \triangleq \mathbb{E}_g\left[T_c^{w,h,\epsilon}(g)^k\right]$. We conclude this subsection by collecting some useful bounds for this quantity. As we verify in the supplement, $M_{c,k}^{w,h,\epsilon}$ is continuously differentiable with respect to $\epsilon$ when $\epsilon > 0$. Additionally, we have the following (see supplement):

**Lemma 4.5.** $\frac{\partial M_{x,k}^{w,h,\epsilon}}{\partial h} = \frac{k}{h} M_{x,k}^{w,h,\epsilon}$.

**Lemma 4.6.** *For any $w \ge 0$, $c, h, h' \in \mathbb{R}$, $\epsilon' \ge \epsilon \ge 0$, and even $k \in \mathbb{N}$, we have the bound $|M_{c,k}^{w,h',\epsilon'} - M_{c,k}^{w,h,\epsilon}| \le h^k \left(|(h'/h)^k - 1| + \epsilon' - \epsilon\right)$. In particular, this implies that $\left|\frac{\partial M_{c,k}^{w,h,\epsilon}}{\partial \epsilon}\right| \le h^k$.*

### 4.3  ODE-Driven Perturbation

Denote the parameters of the function constructed in Lemma 4.4 by $\{(h_i(0), w_i, c_i)\}_{1 \le i < m}$. We also define $\epsilon(0)$ to be some arbitrarily small positive quantity satisfying $\epsilon(0) \le \bar{\epsilon}$ for $\bar{\epsilon}$ from Lemma 4.4.

We will design an ODE whose solution specifies a one-parameter family of functions

$$f_t \triangleq \sum_{i=1}^{m-1} T_{c_i}^{w_i, h_i(t), \epsilon(t)} \tag{1}$$

that arise from gradually perturbing the function from Lemma 4.4. Roughly speaking, starting at $h_i(0)$ and $\epsilon(0)$ for all $1 \le i < m$, perturbing the function along this one-parameter family will correspond to keeping the widths $w_i$ and centers $c_i$ of the bumps fixed, increasing the $\epsilon$ parameter of every bump at unit speed, and evolving the heights $h_i(t)$ in such a way that the moments of the pushforward of $\mathcal{N}(0,1)$ under $f_t$ remain constant in $t$ for all $0 \le t \le T$. We illustrate this evolution

in Figure 2. Here $T$ is some horizon which is at least inverse-polynomially large but smaller than $m^{-3/2}$ so that the "edges" $c_i \pm (\epsilon(t) + w_i)$ of the bumps don't collide with each other (this is where we make crucial use of Part 1 of Lemma 4.4). At the end of this horizon, we want to show that the heights will not have changed too much, whereas the bumps now have $\epsilon$ parameter given by inverse-polynomially large $T$. This will imply that $f_T$ has polynomially bounded slopes as desired.

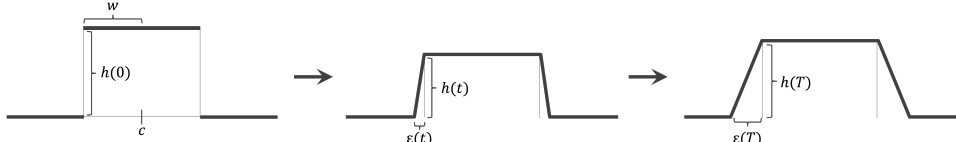

Figure 2: Evolution of one of the $m - 1$ bumps constituting $f_t$

As the odd moments of $f_0(\mathcal{N}(0, 1))$ vanish and the parameters $\{(h_i(0), w_i, c_i)\}_{1 \leq i < m}$ satisfy the symmetry properties from Part 3 of Lemma 4.4, it is easy to ensure that the odd moments of $f_t(\mathcal{N}(0, 1))$ also vanish: simply take $h_i(t) = -h_{m-i}(t)$ for all $1 \leq i \leq m - 1$.

We thus focus on evolving $(h_1(t), \ldots, h_{(m-1)/2}(t))$. For convenience denote this by $\mathbf{h}(t)$. Define the *moment vector* $\mu : \mathbb{R}^{(m-1)/2} \times \mathbb{R} \to \mathbb{R}^{m-1}$ by

$$\mu(\mathbf{h}, \epsilon) \triangleq \left\{ \sum_{i=1}^{(m-1)/2} M_{c_i, 2\ell}^{w_i, h_i, \epsilon} \right\}_{1 \leq \ell \leq (m-1)/2}$$

For any fixed $(\mathbf{h}, \epsilon)$ in a small neighborhood of $(\mathbf{h}(0), \epsilon(0))$, we want to show there is a direction $v \in \mathbb{R}^{(m-1)/2}$ such that the directional derivative of $\mu$ in the direction $w \triangleq (v_1, \ldots, v_{(m-1)/2}, 1)$ is zero. The constraint that the directional derivative $\nabla_w \mu(\mathbf{h}, \epsilon)$ vanishes specifies a linear system in $v$:

$$\sum_{i=1}^{(m-1)/2} v_i \cdot \frac{\partial M_{c_i, 2\ell}^{w_i, h_i, \epsilon}}{\partial h_i} = - \sum_{i=1}^{(m-1)/2} \frac{\partial M_{c_i, 2\ell}^{w_i, h_i, \epsilon}}{\partial \epsilon} \quad \forall \, 1 \leq \ell \leq (m-1)/2. \tag{2}$$

Recalling Lemma 4.5, we can rewrite this as

$$\sum_{i=1}^{(m-1)/2} v_i \cdot \frac{2\ell}{h_i} M_{c_i, 2k}^{w_i, h_i, \epsilon} = - \sum_{i=1}^{(m-1)/2} \frac{\partial M_{c_i, 2\ell}^{w_i, h_i, \epsilon}}{\partial \epsilon} \quad \forall \, 1 \leq \ell \leq (m-1)/2.$$

To express this more compactly, define $\mathbf{b}(\epsilon) \in \mathbb{R}^{(m-1)/2}$ and $Z(\mathbf{h}, \epsilon)$ by

$$\mathbf{b}(\epsilon)_\ell \triangleq - \sum_{i=1}^{(m-1)/2} \frac{\partial M_{c_i, 2\ell}^{w_i, h_i, \epsilon}}{\partial \epsilon} \qquad \text{and} \qquad Z(\mathbf{h}, \epsilon)_{i, \ell} \triangleq M_{c_i, 2\ell}^{w_i, h_i, \epsilon_i}. \tag{3}$$

Also define the matrices $A(\mathbf{h}) \triangleq \mathrm{diag}(1/h_1, \ldots, 1/h_{(m-1)/2})$ and $B \triangleq \mathrm{diag}(2, 4, \ldots, m-1)$. Then (2) is equivalent to $v^\top \cdot A(\mathbf{h}) Z(\mathbf{h}, \epsilon) B = \mathbf{b}(\epsilon)^\top$. Provided $A(\mathbf{h}) Z(\mathbf{h}, \epsilon) B$ is invertible, the natural choice for $v$ would thus be $v = B^{-1} Z(\mathbf{h}, \epsilon)^{-\top} A(\mathbf{h})^{-1} \cdot \mathbf{b}(\epsilon)$. Therefore, defining

$$w(t, \mathbf{h}) \triangleq \left( B^{-1} Z(\mathbf{h}, \epsilon(0) + t)^{-\top} A(\mathbf{h})^{-1} \cdot \mathbf{b}(\epsilon(0) + t), 1 \right), \tag{4}$$

we consider the following initial value problem

$$\mathbf{h}'(t) = w(t, \mathbf{h}(t)) \qquad \text{and} \qquad \mathbf{h}(0) = (h_1(0), \ldots, h_{(m-1)/2}(0)). \tag{5}$$

Note that if we had a solution $\mathbf{h}(t)$ to (5) for $t \in [0, T]$ for some horizon $T$, then we would have

$$\frac{\partial}{\partial t} \mu(\mathbf{h}(t), t) = \left( \frac{\partial}{\partial (\mathbf{h}(t), t)} \mu(\mathbf{h}(t), t) \right) \cdot \mathbf{h}'(t) = \nabla_{w(t)} \mu(\mathbf{h}(t), t) = \mathbf{0}, \tag{6}$$

implying that the low-degree moments of $f_t$ defined in (1) are constant in $t$ as desired.

## 4.4 Existence and Boundedness of $\mathbf{h}(t)$

To carry out the strategy in Section 4.3, we must establish that (1) a solution to the initial value problem (5) exists over a non-negligible horizon $T$, and (2) the entries of $\mathbf{h}(t)$ do not explode in $t$.

For both of these, we need to show that the matrix $Z(\mathbf{h}, \epsilon)$ is invertible or, more specifically, well-conditioned for $(\mathbf{h}, \epsilon)$ in a neighborhood of $(\mathbf{h}(0), \epsilon(0))$. We first establish this at time $t = 0$ by relating $Z(\mathbf{h}(0), \epsilon(0))$ to a certain Vandermonde matrix and appealing to Fact 2.1, see supplement:

**Lemma 4.7.** $\sigma_{\min}(Z(\mathbf{h}(0), \epsilon(0))) \geq m^{-Cm}$ for an absolute constant $C > 0$.

Thus, for $(\mathbf{h}, \epsilon)$ in a neighborhood of $(\mathbf{h}(0), \epsilon(0))$, $Z(\mathbf{h}, \epsilon)$ is well-conditioned (see supplement).

**Lemma 4.8.** Let $C > 0$ be from Lemma 4.7. For any $(\mathbf{h}, \epsilon)$ satisfying $\|\mathbf{h} - \mathbf{h}(0)\|_\infty \leq m^{-C'm}$ and $0 \leq \epsilon - \epsilon(0) \leq m^{-C'm}$ for sufficiently large absolute constant $C' > 0$, $\sigma_{\min}(Z(\mathbf{h}, \epsilon)) \geq m^{-Cm}/2$.

To establish Property (1), we must first verify that $w(t, \mathbf{h})$ is continuous (see supplement for proof):

**Lemma 4.9.** Let $C' > 0$ be the absolute constant from Lemma 4.8. Then the function $w(t, \mathbf{h})$ defined in (4) is continuous with respect to both $t$ and $\mathbf{h}$ for $t \leq m^{-C'm}$ and $\|\mathbf{h} - \mathbf{h}(0)\|_\infty \leq m^{-C'm}$.

Lastly, we must show that under the hypotheses of Lemma 4.9, $\|w(t, \mathbf{h})\|_\infty$ is not too large. This implies Property (1) by Theorem 2.2 and Property (2) because $\|\mathbf{h}'(t)\|_\infty = \|w(t, \mathbf{h}(t))\|_\infty$:

**Lemma 4.10.** Let $C' > 0$ be the absolute constant from Lemma 4.8. If $t \leq m^{-C'm}$ and $\|\mathbf{h} - \mathbf{h}(0)\|_\infty \leq m^{-C'm}$, then $\|w(t, \mathbf{h})\|_\infty \leq m^{C''m}$ for some absolute constant $C'' > 0$.

*Proof.* By the second part of Lemma 4.6, every entry of $\mathbf{b}(\epsilon(0)+t)$ is at most $\frac{m-1}{2} \cdot (\max_i h_i)^{m-1} \leq \frac{m-1}{2} \cdot (O(\sqrt{m}) + m^{-C'm})^{m-1} \leq m^{O(m)}$, where in the penultimate step we used Part 4 of Lemma 4.4 and our hypothesis on $\mathbf{h}$. Note that $\sigma_{\min}(Z(\mathbf{h}, \epsilon(0) + t)) \geq m^{-Cm}/2$ by Lemma 4.8, $\sigma_{\min}(B) \geq 2$, and $\sigma_{\min}(A(\mathbf{h})) \geq \min_i 1/h_i \geq \Omega(\sqrt{m})$ by Part 4 of Lemma 4.4 and our hypothesis on $\mathbf{h}$. We conclude that $B^{-1}Z(\mathbf{h}, \epsilon(0) + t)^{-\top} A(\mathbf{h})^{-1} \cdot \mathbf{b}(\epsilon(0) + t)$ has $L_\infty$ norm at most $m^{C''m}$ for some absolute constant $C'' > 0$, so $\|w(t, \mathbf{h})\|_\infty \leq m^{C''m}$ as claimed. $\square$

We are now ready to put these ingredients together to prove the key lemma for showing Theorem 3.2:

**Lemma 4.11.** Fix any odd $m \in \mathbb{N}$ and any $0 < \nu < 1$. There is a one-hidden-layer ReLU network $f : \mathbb{R} \to \mathbb{R}$ of size $O(m)$ with weights at most $m^{O(m)}$ for which the pushforward $D = f(\mathcal{N}(0, 1))$ satisfies $|\mathbb{E}_{x \sim D}[x^k] - \mathbb{E}_{g \sim \mathcal{N}(0,1)}[g^k]| < \nu$ for all $k = 1, \dots, m$.

*Proof.* Define the parallelepiped $B$ of pairs $(t, \mathbf{h})$ for which $0 \leq t \leq m^{-C'm}$ and $\|\mathbf{h} - \mathbf{h}(0)\|_\infty \leq m^{-C'm}$. By Lemma 4.9, $w$ is continuous over $B$. By Lemma 4.10, $\|w(t, \mathbf{h})\|_\infty \leq m^{C''m}$.

By Theorem 2.2, the initial value problem in (5) has a solution $\mathbf{h}(t)$ over $t \in [0, T]$ for $T = m^{-(C'+C'')m}$. Furthermore, because $\|\mathbf{h}'(t)\|_\infty = \|w(t, \mathbf{h}(t))\|_\infty \leq m^{C''m}$, we conclude that $\frac{1}{T}\|\mathbf{h}(T)\|_\infty \leq m^{C''m}$. The slopes of the bumps $T_{c_i}^{w_i, h_i(T), \epsilon(0)+T}$ are therefore bounded by $m^{C''m}$.

For $(m-1)/2 < i \leq m - 1$, define $h_i(T) = -h_{m-i}(T)$ and consider the one-parameter family of functions $f_t \triangleq \sum_{i=1}^{m-1} T_{c_i}^{w_i, h_i(t), \epsilon(0)+t}$. Because $c_i = c_{m-i}$ and $w_i = w_{m-i}$ for all $1 \leq i < m$ by Part 3 of Lemma 4.4, we conclude that the odd moments of $f_T(\mathcal{N}(0, 1))$ all vanish. As for the even moments, because $\frac{\partial}{\partial t}\mu(\mathbf{h}(t), t) = \mathbf{0}$ by (6), we conclude that the even moments of $f_T(\mathcal{N}(0, 1))$ up to degree $m - 1$ agree with those of $f_0(\mathcal{N}(0, 1))$. So because $D_0 = f_0(\mathcal{N}(0, 1))$ satisfies $|\mathbb{E}_{x \sim D}[x^k] - \mathbb{E}_{g \sim \mathcal{N}(0,1)}[g^k]| < \nu$ for $1 \leq k \leq m$ by Part 2 of Lemma 4.4, the same holds for $f_T$.

As the endpoints of the intervals supporting the bumps are bounded by $O(\log m)$, Fact 4.3 implies $f_T$ is a one-hidden-layer network with size $O(m)$ and $m^{O(m)}$-bounded weights. $\square$

## 4.5 Proof of Theorem 3.2

Having designed a pushforward given by a one-hidden-layer network with $m^{O(m)}$-bounded weights satisfying the first part of Theorem 3.2, we now modify this to also satisfy the remaining two parts.

To do this, we convolve a suitable scaling of the pushforward by a thin Gaussian. The following lemmas show that the moments continue to match, the chi-squared divergence between the result and $\mathcal{N}(0, 1)$ is bounded, and the third part of Theorem 3.2 is satisfied (see supplement for proof):

**Lemma 4.12.** *Let $D$ be any symmetric distribution for which $|\mathbb{E}_{x \sim D}[x^k] - \mathbb{E}_g[g^k]| < \nu$ for all $1 \le k < m$. For any $c \in \mathbb{R}$ let $c \cdot D$ denote the distribution obtained by rescaling $D$ by a factor of $c$. Then $D' \triangleq \sqrt{1 - \sigma^2} \cdot D \star \mathcal{N}(0, \sigma^2)$ satisfies $|\mathbb{E}_{x \sim D'}[x^k] - \mathbb{E}_g[g^k]| < \nu$ for all $1 \le k < m$.*

**Lemma 4.13.** *For any density $A$ on $[-R, R]$ and $\sigma \le 1/2$, $\chi^2(A \star \mathcal{N}(0, \sigma^2), \mathcal{N}(0, 1)) \le e^{O(R^2)}/\sigma$.*

**Lemma 4.14.** *Let $D = f(\mathcal{N}(0, 1))$ be from Lemma 4.11, and define $D' \triangleq \sqrt{1 - \sigma^2} \cdot D \star \mathcal{N}(0, \sigma^2)$. Then for any $v, v' \in \mathbb{S}^{d-1}$ satisfying $|\langle v, v' \rangle| \le 1/2$, $d_{\mathrm{TV}}(P_v^{D'}, P_{v'}^{D'}) \ge 1 - 2\sigma \log(1/\sigma) - m^{-\Omega(m)}$.*

*Proof of Theorem 3.2.* Let $f$ be the function constructed in Lemma 4.11 and define $f^* : \mathbb{R}^2 \to \mathbb{R}$ by $f^*(z_1, z_2) = \sqrt{1 - \sigma^2} f(z_1) + \sigma z_2$. Note that $f^*(\mathcal{N}(0, \mathrm{Id}))$ is exactly $\sqrt{1 - \sigma^2} f(\mathcal{N}(0, 1)) \star \mathcal{N}(0, \sigma^2)$, so the three parts of Theorem 3.2 follow immediately from Lemmas 4.12, 4.13, and 4.14 respectively. □

**Conclusion.** In this work we established that learning one-hidden-layer ReLU network pushforwards of logarithmic size is hard for efficient SQ algorithms. Given that SQ algorithms capture a large family of efficient algorithms, this suggests that this basic learning problem, despite its simple description, is computationally intractable. Of course, our lower bound construction is worst case, and so it is unlikely to actually arise in practice. A natural open question, as mentioned previously, is to come up with natural assumptions (for instance, smoothed analysis settings like in [CLLZ22]) that allow us to circumvent this lower bound, and to achieve efficient learning algorithms.

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
