**Roadmap.** In Section A we provide the deferred proofs from Section 3. In Section B we provide the deferred proofs from Section 4. In Section C we show how to extend our main lower bound to apply to learning in Wasserstein distance. In Section D we fill in the details alluded to in the introduction for how supervised learning lower bounds can imply unsupervised learning lower bounds.

# A Deferred Proofs From Section 3

We begin by reviewing standard concepts pertaining to establishing statistical query lower bounds for unsupervised learning problems, as developed in [FGR$^+$17].

**Definition 4** (Distributional search problems). *Let $\mathcal{D}$ be a set of probability distributions, let $\mathcal{F}$ be a set of* solutions, *and let $\mathcal{Z} : \mathcal{D} \to 2^{\mathcal{F}}$ be a map that takes any $D \in \mathcal{D}$ to a subset of $\mathcal{F}$ corresponding to the valid solutions for $D$. We say that $\mathcal{Z}$ specifies a* distributional search problem over $\mathcal{D}$ and $\mathcal{F}$: *given oracle access to an unknown $D \in \mathcal{D}$, the goal of the learner is to output a valid solution from $\mathcal{Z}(D)$.*

**Definition 5** (Statistical query oracles). *Given a distribution $D$ over $\mathbb{R}^d$ and parameters $\tau, t > 0$, a $\mathrm{STAT}(\tau)$ oracle takes in any query of the form $f : \mathbb{R}^d \to [-1, 1]$ and outputs a value from the interval $[\mathbb{E}_{x \sim D}[f(x)] - \tau, \mathbb{E}_{x \sim D}[f(x)] + \tau]$, while a $\mathrm{VSTAT}(t)$ oracle takes in any query of the form $f : \mathbb{R}^d \to [0, 1]$ and outputs a value from the interval $[\mathbb{E}_{x \sim D}[f(x)] - \tau, \mathbb{E}_{x \sim D}[f(x)] + \tau]$ for $\tau = \max(1/t, \sqrt{\mathbb{V}_{x \sim D}[f(x)]/t})$.*

**Definition 6** (Pairwise correlation). *Given distributions $p, q$ over a domain $\Omega$ which are absolutely continuous with respect to a distribution $r$ over $\Omega$, we let $\chi_r^2(p, q)$ denote the* pairwise correlation, *that is*

$$\chi_r^2(p, q) \triangleq \int_\Omega p(x)q(x)/r(x) \, \mathrm{d}x - 1.$$

*Note that when $p = q$, this is simply the chi-squared divergence between $p$ and $r$.*

*We say that a set of $m$ distributions $\mathcal{D} = \{D_1, \ldots, D_m\}$ is $(\gamma, \beta)$-correlated relative to a distribution $\mu$ over $\mathbb{R}^d$ if*

$$|\chi_\mu(D_i, D_j)| \leq \begin{cases} \gamma & \text{if } i \neq j \\ \beta & \text{if } i = j \end{cases}.$$

**Definition 7** (Statistical dimension). *Let $\beta, \gamma > 0$, let $\mathcal{Z}$ be a distributional search problem over distributions $\mathcal{D}$ and solutions $\mathcal{F}$, and let $N$ be the largest integer for which there exists a distribution $\mu$ and a finite subset $\mathcal{D}_\mu \subseteq \mathcal{D}$ such that for any $f \in \mathcal{F}$, $\mathcal{D}_f \triangleq \mathcal{D}_\mu \backslash \mathcal{Z}^{-1}(f)$ is $(\gamma, \beta)$-correlated relative to $\mu$ and $|\mathcal{D}_f| \geq N$. We say that the* statistical dimension *with pairwise correlations $(\gamma, \beta)$ of $\mathcal{Z}$ is $N$ and denote it by $\mathrm{SD}(\mathcal{Z}, \gamma, \beta)$.*

**Lemma A.1** (Corollary 3.12 from [FGR$^+$17]). *Let $\mathcal{Z}$ be a distributional search problem over distributions $\mathcal{D}$ and solutions $\mathcal{F}$. For $\gamma, \beta$, if $N = \mathrm{SD}(\mathcal{Z}, \gamma, \beta)$, then any statistical query algorithm for $\mathcal{Z}$ requires at least $N\gamma/(\beta - \gamma)$ queries to $\mathrm{STAT}(\sqrt{2\gamma})$ or $\mathrm{VSTAT}(1/(6\gamma))$.*

We will use the following two lemmas from [DKS17] and [DK20]. Recall from the main body that $P_v^D$ denotes the distribution over $\mathbb{R}^d$ with density

$$P_v^D(x) = D(\langle v, x \rangle) \cdot \gamma^{(d-1)}(x - \langle v, x \rangle v),$$

that is the distribution which is given by $D$ in the direction $v$ and is given by $\mathcal{N}(0, \mathrm{Id} - vv^\top)$ orthogonal to $v$.

**Lemma A.2** (Lemma 3.5 from [DK20]). *There is an absolute constant $c > 0$ such that the following holds. Let $m \in \mathbb{N}$ and $\nu > 0$. If a distribution $D$ over $\mathbb{R}$ is such that 1) $\chi^2(D, \mathcal{N}(0, 1))$ is finite, and 2) $|\mathbb{E}_{x \sim D}[x^k] - \mathbb{E}_{g \sim \mathcal{N}(0, 1)}[g^k]| \leq \nu$ for all $k = 1, \ldots, m$, then for all $v, v' \in \mathbb{S}^{d-1}$ for which $|\langle v, v' \rangle| < c$,*

$$|\chi_{\mathcal{N}(0, \mathrm{Id}_d)}^2(P_v^D, P_{v'}^D)| \leq |\langle v, v' \rangle|^{m+1} \chi^2(D, \mathcal{N}(0, 1)) + \nu^2.$$

**Fact A.3** (Lemma 3.7 from [DKS17]). *For any constant $0 < C < 1/2$, there exists a set $S$ of $2^{d^C}$ unit vectors in $\mathbb{S}^{d-1}$ such that any pair of distinct $u, v \in S$ satisfies $|\langle u, v \rangle| < d^{C-1/2}$.*

We can now prove the generic statistical query lower bound of Lemma 3.3:

*Proof of Lemma 3.3.* Let $S$ be the set of $2^{d^C}$ unit vectors from Fact A.3. In the notation of Definition 7, take $\mu = \mathcal{N}(0, \mathrm{Id})$ and let $\mathcal{D}_\mu \triangleq \{P_v^D\}_{v \in S}$. By Lemma A.2, for any distinct $v, v' \in S$ we have

$$\chi_\mu(P_v^D, P_{v'}^D) \le |\langle v, v' \rangle|^{m+1} \chi^2(D, \mathcal{N}(0,1)) + O(\tau^2) \le \Omega(d)^{-(m+1)(1/2-C)} \chi^2(D, \mathcal{N}(0,1)).$$

On the other hand, if $v = v' \in S$, then $\chi_\mu(P_v^D, P_v^D) = \chi^2(D, \mathcal{N}(0,1)) + O(\tau^2) \le 2\chi^2(D, \mathcal{N}(0,1))$. So for

$$\gamma \triangleq \Omega(d)^{-(m+1)(1/2-C)} \chi^2(D, \mathcal{N}(0,1)) \qquad \text{and} \qquad \beta \triangleq 2\chi^2(D, \mathcal{N}(0,1)),$$

$\mathcal{D}_\mu$ is $(\gamma, \beta)$-correlated with respect to $\mu$.

Consider the distributional search problem $\mathcal{Z}$ mapping any distribution $P_v^D$ to the set of probability distributions which are $\epsilon$-close in total variation distance to $P_v^D$. Because $\mathrm{d}_{\mathrm{TV}}(P_v, P_{v'}) > 2\epsilon$ for distinct $v, v' \in S$, for any distribution $f$ over $\mathbb{R}^d$ we have that $|\mathcal{Z}^{-1}(f)| \le 1$. We conclude that $\mathrm{SD}(\mathcal{Z}, \gamma, \beta) \ge 2^{\Omega(d^C)}$. By Lemma A.1, we conclude that any SQ algorithm for $\mathcal{Z}$ requires at least $2^{\Omega(d^C)} d^{-(m+1)(1/2-C)}$ calls to either $\mathrm{STAT}(\tau)$ or $\mathrm{VSTAT}(1/\tau^2)$. Note that because we are assuming that $d \ge m^{\Omega(1/C)}$, we have $2^{\Omega(d^{C/2})} \ge d^{m+1}$, so the total number of required queries is at least $2^{\Omega(d^{C/2})} \ge d^{m+1}$ as claimed. $\qquad\square$

# B    Deferred Proofs From Section 4

## B.1    Proof of Fact 4.3

*Proof.* We have for all $z \in \mathbb{R}$ that $T_c^{w,h,\epsilon}(z)$ is equal to

$$\frac{h}{\epsilon} \left( \mathrm{ReLU}(z - c + \epsilon + w) - \mathrm{ReLU}(z - c + w) - \mathrm{ReLU}(z - c - w) + \mathrm{ReLU}(z - c - \epsilon - w) \right).$$

$\square$

## B.2    Proof of Lemma 4.4

To prove Lemma 4.4, we will need the following modification of the construction in Lemma 4.1.

**Lemma B.1.** *For any odd $m \in \mathbb{N}$, there exist weights $\lambda_1, \ldots, \lambda_{m-1} \ge 0$ and points $h_1, \ldots, h_{m-1} \in \mathbb{R}$ for which*

1. *(Sum of weights bounded away from 1)* $\sum_{i=1}^{m-1} \lambda_i = 1 - \Theta(1/\sqrt{m})$.

2. *(Moments match)* $|\sum_{i=1}^{m-1} \lambda_i h_i^k - \mathbb{E}_g[g^k]| < \nu$ *for all $k = 1, \ldots, 2m - 1$.*

3. *(Points symmetric about origin)* $h_1 \le \cdots \le h_{m-1}$ *and $h_i = -h_{m-i}$ for all $1 \le i < m$.*

4. *(Weights symmetric)* $\lambda_1 \le \cdots \le \lambda_{(m-1)/2}$ *and $\lambda_i = \lambda_{m-i}$.*

5. *(Points bounded and separated)* $\Omega(1/\sqrt{m}) \le |h_i| \le O(\sqrt{m})$ *for all $1 \le i < m$, and $\{h_i\}$ are $\Omega(1/\sqrt{m})$-separated.*

6. *(Weights not too small)* $\min_i \lambda_i \ge e^{-cm}$ *for an absolute constant $c > 0$.*

*Proof.* Because $m$ is odd, we can take the weights and points to be given by Lemma 4.1 and remove the $(m+1)/2$-th weight and point– recall that the $(m+1)/2$-th point is 0 and thus does not contribute to $\sum_i \lambda_i h_i^k$. The fact that $\sum_{i=1}^{m-1} \lambda_i = 1 - \Theta(1/\sqrt{m})$ then follows from the fact that the $(m+1)/2$-th weight from Lemma 4.1 is of order $\Theta(1/\sqrt{m})$ by Part 6 of Lemma 4.1. The remaining parts of the lemma follow by the corresponding parts of Lemma 4.1. $\qquad\square$

*Proof of Lemma 4.4.* Let $\lambda_1, \ldots, \lambda_{m-1}, h_1, \ldots, h_{m-1}$ be as in Lemma B.1. As $\sum_i \lambda_i = 1 - \Theta(1/\sqrt{m})$, we claim there exist intervals $I_1, \ldots, I_{m-1}$ such that for any $i < j$, all points in $I_i$ are strictly smaller than all points in $I_j$, such that $\gamma(I_i) = \lambda_i$ for all $i$, and such that the right endpoint of any $I_i$ is at least $m^{-3/2}$ smaller than the left endpoint of $I_{i+1}$.

We can construct these intervals in an inductive fashion. First, let $\gamma \triangleq 1 - \sum_i \lambda_i = \Theta(1/\sqrt{m})$. Let $I_1 = [a_1, b_1]$ for $a_1 < b_1 < 0$ such that $\gamma(I_1) = \lambda_1$ and $\gamma((-\infty, a_1]) = \gamma/m$. Given $I_1, \ldots, I_i$ for $1 \le i < (m-1)/2$, if $I_i = [a_i, b_i]$ is the right endpoint of $I_i$, then let $a_{i+1} > b_i$ be such that $\gamma([b_i, a_{i+1}]) = \gamma/m$, and define $I_{i+1} = [a_{i+1}, b_{i+1}]$ for $b_{i+1} > a_{i+1}$ satisfying $\gamma([a_{i+1}, b_{i+1}]) = \lambda_{i+1}$. By construction,

$$\gamma((-\infty, b_{(m-1)/2}]) = \sum_{i=1}^{(m-1)/2} \gamma(I_i) + \frac{m-1}{2} \cdot \frac{\gamma}{m} = \frac{1-\gamma}{2} + \frac{m-1}{2m} \cdot \frac{\gamma}{2} = \frac{1}{2} - \frac{\gamma}{2m},$$

so by Gaussian anticoncentration and the fact that $\gamma = \Theta(1/\sqrt{m})$, we conclude that $b_{(m-1)/2} \le -\Omega(m^{-3/2})$. In the same way, we also conclude that because $\gamma([b_i, a_{i+1}]) = \frac{\gamma}{m}$, we must have $b_i - a_{i+1} \le -\Omega(m^{-3/2})$. Finally, for $i = (m-1)/2 + 1, \ldots, m-1$, we can define $I_i$ to be the reflection of $I_{m-i}$ about the origin. Note that by our bounds on $b_i - a_{i+1}$ for $1 \le i \le (m-1)/2$ and on $b_{(m-1)/2}$, all of the intervals are $\Omega(m^{-3/2})$-separated from each other as claimed. And by design, $\gamma(I_i) = \lambda_i$ for all $1 \le i \le m-1$.

While the lemma is stated in terms of $\epsilon > 0$, let us first consider the following construction where $\epsilon = 0$. We can take the centers $c_1, \ldots, c_{m-1}$ in the lemma to be the centers of $I_1, \ldots, I_{m-1}$, and $w_1, \ldots, w_{m-1}$ to be half of the widths of $I_1, \ldots, I_{m-1}$, in which case $f \triangleq \sum_{i=1}^{m-1} T_{c_i}^{w_i, h_i, 0}$ immediately satisfies Parts 1 and 3 of the lemma. Then the pushforward of $\mathcal{N}(0, 1)$ under this choice of $f$ is the distribution which with probability $\gamma$ equals zero (when $z \sim \mathcal{N}(0, 1)$ lies outside of $I_1, \ldots, I_{m-1}$) and otherwise takes the value $h_i$ with probability $\lambda_i$. Parts 2, 4, and 5 then follow from Lemma B.1. Finally, note that $a_1$ defined above is at most $O(\log m)$ in magnitude because $\gamma((-\infty, a_1]) = \gamma/m$ by Part 6 of Lemma B.1. This establishes Part 6 of the lemma.

Finally, note that by taking $\epsilon$ infinitesimally small (relative to $\nu$) but positive, the function $f$ defined in the lemma satisfies all of the parts of the lemma. $\qquad\square$

## B.3  Proofs from Section 4.2.1

First, we give an explicit expression for $M_{c,k}^{w,h,\epsilon} \triangleq \mathbb{E}_g[T_c^{w,h,\epsilon}(g)^k]$:

**Lemma B.2.** *For $c, w, h, \epsilon > 0$ satisfying $c - \epsilon - w \ge 0$, we have*

$$M_{c,k}^{w,h,\epsilon} = h^k \gamma([c-w, c+w])+$$

$$(k-1)!! \left(\frac{h}{\epsilon}\right)^k \sum_{\substack{i=0 \\ even}}^{k} \binom{k}{i}\left[(-c+\epsilon+w)^{k-i}\gamma([c-\epsilon-w, c-w]) + (c+\epsilon+w)^{k-i}\gamma([c+w, c+\epsilon+w])\right]$$

$$-\left(\frac{h}{\epsilon}\right)^k \sum_{i=0}^{k} \binom{k}{i}\left[(-c+\epsilon+w)^{k-i}\left(p_i(c-w)\gamma(c-w) - p_i(c-\epsilon-w)\gamma(c-\epsilon-w)\right)\right.$$

$$\left. + (-1)^i(c+\epsilon+w)^{k-i}\left(p_i(c+\epsilon+w)\gamma(c+\epsilon+w) - p_i(c+w)\gamma(c+w)\right)\right]$$

*for all even $k$.*

To show this we use the form of the moments of a truncated Gaussian. Given $m, i \in \mathbb{N}$, let $m^{\Downarrow i} \triangleq m(m-2)\cdots(m-2i+2)$. Also let $m^{\Downarrow 0} = 1$. Then:

**Lemma B.3.** *For any $k \in \mathbb{N}$, define the polynomial*

$$p_k(x) \triangleq \sum_{i=0}^{\lfloor (k-1)/2 \rfloor} (k-1)^{\Downarrow i} x^{k-1-2i}.$$

*For any $a \le b$,*

$$\mathbb{E}_g[g^k \cdot \mathbb{1}[a \le g \le b]] = \begin{cases} (k-1)!! \cdot \gamma([a,b]) - (p_k(b)\gamma(b) - p_k(a)\gamma(a)) & \text{if } k \text{ even} \\ -(p_k(b)\gamma(b) - p_k(a)\gamma(b)) & \text{if } k \text{ odd} \end{cases}$$

**Corollary B.4.** *For any* $c, d \in \mathbb{R}$ *and* $k \in \mathbb{N}$ *even,*

$$\mathbb{E}_g[(cg + d)^k \cdot \mathbb{1}[a \le g \le b]] = \sum_{\substack{i=0 \\ even}}^{k} \binom{k}{i} c^i d^{k-i}(k-1)!!\gamma([a,b])$$

$$- \sum_{i=0}^{k} \binom{k}{i} c^i d^{k-i}(p_i(b)\gamma(b) - p_i(a)\gamma(a)).$$

*Proof of Lemma B.2.* By Corollary B.4, the contribution from the interval $g \in [c - \epsilon - w, c - w]$ to $\mathbb{E}_g\left[\left(T_c^{w,h,\epsilon}(z)\right)^k\right]$ is given by

$$\sum_{\substack{i=0 \\ even}}^{k} \binom{k}{i} \left(\frac{h}{\epsilon}\right)^i \left(\frac{h}{\epsilon}(-c + \epsilon + w)\right)^{k-i} (k-1)!!\gamma([c - \epsilon - w, c - w])-$$

$$\sum_{i=0}^{k} \binom{k}{i} \left(\frac{h}{\epsilon}\right)^i \left(\frac{h}{\epsilon}(-c + \epsilon + w)\right)^{k-i} (p_i(c - w)\gamma(c - w) - p_i(c - \epsilon - w)\gamma(c - \epsilon - w)).$$

Similarly, the contribution from the interval $g \in [c + w, c + \epsilon + w]$ is given by

$$\sum_{\substack{i=0 \\ even}}^{k} \binom{k}{i} \left(\frac{-h}{\epsilon}\right)^i \left(\frac{h}{\epsilon}(c + \epsilon + w)\right)^{k-i} (k-1)!!\gamma([c + w, c + \epsilon + w])-$$

$$\sum_{i=0}^{k} \binom{k}{i} \left(\frac{-h}{\epsilon}\right)^i \left(\frac{h}{\epsilon}(c + \epsilon + w)\right)^{k-i} (p_i(c + \epsilon + w)\gamma(c + \epsilon + w) - p_i(c + w)\gamma(c + w)).$$

Finally, the contribution from the interval $g \in [c - w, c + w]$ is given by $h^k \cdot \gamma([c - w, c + w])$. $\square$

Lemma 4.5 now follows immediately from LemmaB.2. Furthermore, it is clear from Lemma B.2 that $M_{c,k}^{w,h,\epsilon}$ is continuously differentiable with respect to $\epsilon$ when $\epsilon > 0$.

*Proof of Lemma 4.6.* Note that $0 \le M_{c,k}^{w,h,\epsilon} \le h^k\gamma([c - \epsilon - w, c + \epsilon + w])$, so the first part of the lemma follows by upper bounding $|M_{c,k}^{w,h',\epsilon'} - M_{c,k}^{w,h,\epsilon}|$ by

$$|h'^k - h^k|\gamma([c - \epsilon' - w, c + \epsilon' + w]) + h^k\gamma([c - \epsilon' - w, c - \epsilon - w] \cup [c + \epsilon + w, c + \epsilon' + w])$$

$$\le |h'^k - h^k| + h^k(\epsilon' - \epsilon) = h^k\left(|(h'/h)^k - 1| + \epsilon' - \epsilon\right),$$

where in the last step we used that $\gamma([a, a + \eta]) \le \eta/2$ for any $a \in \mathbb{R}$, $\eta \ge 0$. The second part of the lemma then follows by taking $h = h'$ and $\epsilon' \to \epsilon$. $\square$

## B.4 Proof of Lemma 4.7

*Proof.* For convenience, in this proof we refer to $Z(\mathbf{h}(0), \epsilon(0))$ as $Z$. Note that $Z_{i,\ell} = \gamma([c_i - w_i, c_i + w_i]) \cdot h_i(0)^{2\ell} + \xi_{i,\ell}$ for some $\xi_{i,\ell}$ which can be made arbitrarily small by taking $\epsilon(0)$ to be arbitrarily small. We can thus write $Z = \Lambda H + \Xi$ for $\Lambda = \text{diag}(\lambda_1 h_1(0)^2, \ldots, \lambda_{(m-1)/2}h_{(m-1)/2}(0)^2)$, $H \in \mathbb{R}^{(m-1)/2 \times (m-1)/2}$ given by $H_{i,\ell} = h_i(0)^{2\ell-2}$, and $\Xi$ a matrix consisting of arbitrarily small positive entries. So $\sigma_{\min}(Z) \ge (\min_i \lambda_i h_i(0)^2) \cdot \sigma_{\min}(H) - \xi \ge (e^{-cm}/m) \cdot \sigma_{\min}(H) - \xi$ for arbitrarily small $\xi > 0$, where in the last step we used Parts 4 and 5 of Lemma 4.4.

Finally, note that $H$ is a Vandermonde matrix with nodes $h_1(0)^2, \ldots, h_{(m-1)/2}(0)^2$. As $\{h_i(0)\}$ are $\Omega(1/\sqrt{m})$-separated and lie within $[\Omega(1/\sqrt{m}), O(\sqrt{m})]$, $\{h_i(0)^2\}$ are $\Omega(1/m)$-separated. So by Fact 2.1, $\sigma_{\min}(H) \ge m^{-O(m)}$, concluding the proof. $\square$

## B.5 Proof of Lemma 4.8

*Proof.* For convenience, in this proof we refer to $Z(\mathbf{h}, \epsilon)$ and $Z(\mathbf{h}(0), \epsilon(0))$ by $Z'$ and $Z$ respectively. By Lemma 4.6, each entry of $Z'$ differs from the corresponding entry of $Z$ by at most

$$(\max_i h_i^{m-1}) \cdot \left( \left| (1 + m^{-C'm}/(\min_i h_i))^{m-1} - 1 \right| + m^{-C'm} - \epsilon(0) \right).$$

As $\max_i h_i \leq O(\sqrt{m})$ and $\min_i h_i \geq \Omega(1/\sqrt{m})$ by Part 4 of Lemma 4.4, and $\epsilon(0)$ can is an arbitrarily small positive quantity, the above is at most $m^{-C''m}$ for some absolute constant $C'' > 0$ which is increasing in $C'$. So $\|Z - Z'\|_{\mathsf{op}} \leq \|Z - Z'\|_F \leq (m-1)/2 \cdot m^{-C''m} \ll m^{-Cm}$ provided we take $C'$ sufficiently large. $\qquad\square$

## B.6 Proof of Lemma 4.9

*Proof of Lemma 4.9.* By our expression for $M_{c,k}^{w,h,\epsilon}$ in Lemma B.2 and the definition of $\mathbf{b}(\epsilon)$ in (3), $\mathbf{b}(\epsilon(0)+t)$ is clearly continuous in $t$ whenever $t \geq 0$ (because $\epsilon(0) > 0$). Similarly, $A(\mathbf{h})Z(\mathbf{h}, \epsilon(0)+t)B$ is clearly continuous with respect to $\mathbf{h}$ and $t$ whenever $t \geq 0$ and $h_i \neq 0$ for all $i$. By Lemma 4.8, if $t \leq m^{-C'm}$ and $\|\mathbf{h} - \mathbf{h}(0)\|_\infty \leq m^{-C'm}$ (which additionally implies that $h_i \neq 0$ for all $i$, by Part 4 of Lemma 4.4), then $A(\mathbf{h})Z(\mathbf{h}, \epsilon(0) + t)B$ is invertible. We conclude that for such $t, \mathbf{h}, w$ is continuous. $\qquad\square$

## B.7 Proof of Lemma 4.12

*Proof.* For any $1 \leq k < m$,

$$\mathop{\mathbb{E}}_{x \sim D'}[x^k] = \mathop{\mathbb{E}}_{z \sim \sqrt{1-\sigma^2} \cdot D, g \sim \mathcal{N}(0,1)} [(z + \sigma g)^k] = \mathop{\mathbb{E}}_{g,g' \sim \mathcal{N}(0,1)} [(\sqrt{1-\sigma^2}g' + \sigma g)^k] = \mathop{\mathbb{E}}_{g \sim \mathcal{N}(0,1)}[g^k],$$

where in the second step we used that $\sqrt{1-\sigma^2} \cdot D$ matches the moments of $\mathcal{N}(0, 1-\sigma^2)$ up to degree $m$, and in the last step we used that $\sqrt{1-\sigma^2}g' + \sigma g$ is distributed as a draw from $\mathcal{N}(0,1)$. $\quad\square$

## B.8 Proof of Lemma 4.13

*Proof.* Let $\widetilde{A} \triangleq A \star \mathcal{N}(0,1)$. By definition, $\widetilde{A}(x) = \int_{-\infty}^\infty A(s)\gamma_{\sigma^2}(x-s)\,\mathrm{d}s$. So

$$1 + \chi^2(\widetilde{A}, \mathcal{N}(0,1)) = \int_{-\infty}^\infty \frac{1}{\gamma(x)} \left( \int_{-\infty}^\infty \int_{-\infty}^\infty A(s)A(t)\gamma_{\sigma^2}(x-s)\gamma_{\sigma^2}(x-t)\,\mathrm{d}s\mathrm{d}t \right) \mathrm{d}x. \quad (7)$$

Note that for any $s, t \in \mathbb{R}$,

$$\frac{\gamma_{\sigma^2}(x-s)\gamma_{\sigma^2}(x-t)}{\gamma(x)} = \frac{1}{\sigma^2\sqrt{2\pi}} \exp\left( \frac{-(x-s)^2 - (x-t)^2}{2\sigma^2} + x^2/2 \right)$$

$$= \frac{1}{\sigma^2\sqrt{2\pi}} \exp\left( -\frac{2-\sigma^2}{2\sigma^2}\left( x - \frac{s+t}{2-\sigma^2} \right)^2 + \frac{2st - (s^2+t^2)(1-\sigma^2)}{2\sigma^2(2-\sigma^2)} \right),$$

so

$$\int_{-\infty}^\infty \frac{\gamma_{\sigma^2}(x-s)\gamma_{\sigma^2}(x-t)}{\gamma(x)}\,\mathrm{d}x = \frac{e^{\frac{2st - (s^2+t^2)(1-\sigma^2)}{2\sigma^2(2-\sigma^2)}}}{\sigma\sqrt{2-\sigma^2}}.$$

Eq. (7) thus becomes

$$1 + \chi^2(\widetilde{A}, \mathcal{N}(0,1)) = \int_{-\infty}^\infty \int_{-\infty}^\infty A(s)A(t) \cdot \frac{e^{\frac{2st - (s^2+t^2)(1-\sigma^2)}{2\sigma^2(2-\sigma^2)}}}{\sigma\sqrt{2-\sigma^2}}\,\mathrm{d}s\mathrm{d}t \quad (8)$$

As $A$ is supported on $[-R, R]$,

$$\frac{e^{\frac{2st - (s^2+t^2)(1-\sigma^2)}{2\sigma^2(2-\sigma^2)}}}{\sigma\sqrt{2-\sigma^2}} \leq e^{O(R^2/(1-\sigma^2))} \leq e^{O(R^2)}.$$

Substituting this into (8), we find that

$$1 + \chi^2(\widetilde{A}, \mathcal{N}(0,1)) \leq \frac{e^{O(k)}}{\sigma\sqrt{2-\sigma^2}} \int \int A(s)A(t)\,\mathrm{d}s\mathrm{d}t = \frac{e^{O(R^2)}}{\sigma\sqrt{2-\sigma^2}} \leq e^{O(R^2)}/\sigma. \qquad \square$$

## B.9 Proof of Lemma 4.14

We will use the following elementary fact about total variation distance:

**Fact B.5.** *Given two distributions $p, q$ over a domain $\Omega$, $d_{\mathrm{TV}}(p, q) = 1 - \int_\Omega \min(p(x), q(x))\,\mathrm{d}x$.*

*Proof of Lemma 4.14.* By Fact B.5, it suffices to upper bound $\int_{\mathbb{R}^d} \min(P_v(z), P_{v'}(z))\,\mathrm{d}z$. Let $H$ denote the plane spanned by $v, v'$. As the component in $H$ of a sample from either $P_v$ or $P_{v'}$ is independent from the component in $H^\perp$, and the latter is distributed as $\mathcal{N}(0, \Pi_{H^\perp})$, it suffices to bound $\int_H \min(P_v(z), P_{v'}(z))\,\mathrm{d}z$. Let $x, y$ be orthogonal coordinates for $H$ with $v$ in the direction of the $x$-axis, and let $x', y'$ be orthogonal coordinates for $H$ with $v'$ in the direction of the $x'$-axis. Let $\theta$ be the angle between $v, v'$. Then

$$
\int_H \min(P_v(z), P_{v'}(z))\,\mathrm{d}z = \int_{-\infty}^\infty \int_{-\infty}^\infty \min(D'(x)\gamma(y), D'(x')\gamma(y'))\,\mathrm{d}x\mathrm{d}y
$$

$$
= \int_{-\infty}^\infty \int_{-\infty}^\infty \min(D'(x)\gamma(y), D'(x')\gamma(y'))\csc\theta\,\mathrm{d}x\mathrm{d}x'. \qquad (9)
$$

For $1 \le i \le m-1$, let $D_i$ denote the distribution of $\sqrt{1-\sigma^2} \cdot T_{x_i}^{w_i, \mathbf{h}_i(T), T}(g)$ for $g \sim \mathcal{N}(0, 1)$ conditioned on $g \in [x_i - w_i - T, x_i + w_i + T]$. Also let $D_m$ denote the distribution which is a point mass at zero. Let $D_i' \triangleq D_i \star \mathcal{N}(0, 1)$. Note that there is a distribution $p$ over $[m]$ for which $D' = \mathbb{E}_{i \sim p}[D_i']$. Then we can upper bound (9) by

$$
\max_{i,j \in [m]} \int_{-\infty}^\infty \int_{-\infty}^\infty \min(D'(x)_i\gamma(y), D_j'(x')\gamma(y'))\csc\theta\,\mathrm{d}x\mathrm{d}x' \qquad (10)
$$

Note that for $1 \le i \le m-1$, $\mathbb{P}_{x \sim D_i'}[|x - x_i| > a] \le \sigma + \xi_i$ for $a \triangleq 2\sigma\sqrt{\log(1/\sigma)}$ and $\xi_i \triangleq \frac{\gamma([x_i - w_i - T, x_i - w_i] \cup [x_i + w_i, x_i + w_i + T])}{\gamma([x_i - w_i - T, x_i + w_i + T])}$. And for $i = m$, $\mathbb{P}_{x \sim D_m'}[|x - x_i| > a] \le \sigma$ for $x_m \triangleq 0$. So we get an upper bound on (10) of

$$
\sigma + \max_{i \in [m-1]} \xi_i + \max_{i,j \in [m]} \int_{x_i - a}^{x_i + a} \int_{x_j - a}^{x_j + a} \min(D_i'(x)\gamma(y), D_j'(x')\gamma(y'))\csc\theta\,\mathrm{d}x\mathrm{d}x'
$$

As $D_i'$ is a convolution of $D_i$ with $\mathcal{N}(0, \sigma^2)$, $D_i'(x) \le \frac{1}{\sigma\sqrt{2\pi}}$ for all $x \in \mathbb{R}$. And $\gamma(y) \le 1/\sqrt{2\pi}$, so the above display is at most

$$
\sigma + \max_{i \in [m-1]} \xi_i + a^2\csc\theta/(2\pi\sigma) = \sigma + \max_{i \in [m-1]} \xi_i + \sigma\csc\theta\log(1/\sigma)/\pi.
$$

Note that if $|\langle v, v'\rangle| \le 1/2$, then $\csc\theta/\pi \le 2/(\sqrt{3}\pi) \le 1$. Finally, to bound $\xi_i$, first note that for any $i \in [m-1]$, $\gamma([x_i - w_i - T, x_i + w_i + T]) = \lambda_i$, and recall that $\lambda_i \ge e^{-cm}$ for some absolute constant $c > 0$. On the other hand, $\gamma([x_i - w_i - T, x_i - w_i]) \le T/\sqrt{2\pi} = m^{-(C'+C'')m}/\sqrt{2\pi}$ by Gaussian anti-concentration and our choice of $T = m^{-(C'+C'')m}$ in the proof of Lemma 4.11. So by taking the constants $C', C''$ to be larger than $c$, we conclude that $\xi_i \le m^{-\Omega(m)}$ for all $i \in [m-1]$. $\qquad \square$

## C Hardness for Estimation in Wasserstein

We now show an analogous version of Theorem 3.1 under the Wasserstein-1 metric rather than total variation distance. We begin by observing that the ODE-based evolution does not move the pushforward at time zero, i.e. the distribution constructed in Lemma 4.4, too far away in Wasserstein distance over a time horizon of $T$:

**Lemma C.1.** *Let $f_0, f_T : \mathbb{R} \to \mathbb{R}$ denote the functions from Lemmas 4.4 and 4.11 respectively. Define $D_0 \triangleq f_0(\mathcal{N}(0, 1))$ and $D_T \triangleq f_T(\mathcal{N}(0, 1))$. Then $W_1(D, D') \le m^{-\Omega(m)}$.*

*Proof.* Recall that $w_1, \ldots, w_{m-1}$ denote the widths of the bumps in $f_0, f_T$, $x_1, \ldots, x_{m-1}$ denote the centers, and the heights and $\epsilon$ parameters for the bumps in $f_0, f_T$ are given by $\{h_i(0)\}_i, \epsilon(0)$

and $\{h_i(T)\}_i, \epsilon(0) + T$ respectively, for $T = m^{-(C'+C'')m}$ and $\epsilon(0)$ an arbitrarily small positive quantity. Also recall from the proof of Lemma 4.11 that $|h_i(0) - h_i(T)| \le m^{-C'm}$ for all $i$.

Now consider any $g \in \mathbb{R}$. If $g \in [x_i - w_i, x_i + w_i]$ for some $i$, then $|f_0(g) - f_T(g)| = |h_i(0) - h_i(T)| \le m^{-C'm}$. Furthermore,

$$\mathop{\mathbb{P}}_{g \sim \mathcal{N}(0,1)} [g \in [x_i - \epsilon_i(0) - T - w_i, x_i - w_i] \cup [x_i + w_i, x_i + \epsilon_i(0) + T + w_i] \text{ for some } i] \le O(mT)$$

Finally, for all $g$ that do not lie in any of the aforementioned intervals, i.e. that do not lie in the support of any bump from $f_0$ or $f_T$, note that $f_0(g) = f_T(g) = 0$ by construction. We conclude that for any 1-Lipschitz function $h : \mathbb{R} \to \mathbb{R}$,

$$\left| \mathop{\mathbb{E}}_{z \sim D}[h(z)] - \mathop{\mathbb{E}}_{z \sim D'}[h(z)] \right| = \left| \mathop{\mathbb{E}}_{g}[h(f_0(g)) - h(f_T(g))] \right|$$
$$\le \mathop{\mathbb{E}}_{g}[|f_0(g) - f_T(g)|] \le m^{-C'm} + O(mT) \le m^{-\Omega(m)}$$

as claimed. $\qquad\square$

We can now show the analogue of Lemma 4.14 for Wasserstein distance:

**Lemma C.2.** *Let $D = f(\mathcal{N}(0,1))$ be from Lemma 4.11, and define $D' \triangleq \sqrt{1 - \sigma^2} \cdot D \star \mathcal{N}(0, \sigma^2)$ for $\sigma \ll 1/\sqrt{m}$. Then for any $v, v' \in \mathbb{S}^{d-1}$ satisfying $|\langle v, v' \rangle| \le 1/2$, $W_1(P_v^{D'}, P_{v'}^{D'}) \ge \Omega(1/\sqrt{m})$.*

*Proof.* Let $f_0$ be the function from Lemma 4.4, and let $A$ denote $f_0(\mathcal{N}(0,1))$. We begin by lower bounding $W_1(P_v^A, P_{v'}^A)$, which we will do by showing that with $\Omega(1)$ probability, a sample $x$ from $P_{v''}^A$ will be distance $\Omega(1/\sqrt{m})$ from the support of $P_v^A$. As the distance from a point $x$ to the affine hyperplane $\Lambda_h \triangleq \{z : \langle z, v \rangle = h\}$ is $|\langle v, x \rangle - h|$, if $x$ is of the form $h'v' + v^\perp$ for some $h' \in \mathbb{R}$, then $x$ is at distance

$$\left| h' \langle v', v \rangle + \langle v^\perp, v \rangle - h \right|$$

from the hyperplane. Note that $P_v^A$ is supported on the hyperplanes $\Lambda_{h_1}, \dots, \Lambda_{h_{m-1}}$ for $h_1, \dots, h_{m-1}$ from Lemma 4.4. And for $x \sim P_{v'}^A$, $h'$ takes on the value $h_i$ with probability $\lambda_i$ (where $\{\lambda_i\}$ are also from Lemma 4.4), while $v^\perp$ is an independent draw from $\mathcal{N}(0, \mathrm{Id} - v'v'^\perp)$. We conclude that $h' \langle v', v \rangle + \langle v^\perp, v \rangle$ is distributed as $\mathcal{N}(h'\langle v', v \rangle, 1 - \langle v', v \rangle^2)$. Therefore, the event that $x$ is at distance $\Omega(1/\sqrt{m})$ from the support of $P_v^A$ is equivalent to the event that a sample from $\mathcal{N}(h'\langle v', v \rangle, 1 - \langle v', v \rangle^2)$ is $\Omega(1/\sqrt{m})$-far from every $h_1, \dots, h_{m-1}$. But note that because $h_1, \dots, h_{m-1}$ are $\Omega(1/\sqrt{m})$-separated, there is an absolute constant $c > 0$ such that the union of the balls of radius $c/\sqrt{m}$ around $h_1, \dots, h_{m-1}$ cover at most a constant fraction of the interval $[h'\langle v', v \rangle - 1, h'\langle v', v \rangle + 1]$. Because $1 - \langle v', v \rangle^2 \ge 3/4$, a constant fraction of the mass of $\mathcal{N}(h'\langle v', v \rangle, 1 - \langle v', v \rangle^2)$ is located in this interval, concluding the proof that $W_1(P_v^A, P_{v'}^A) \ge \Omega(1/\sqrt{m})$.

By Lemma C.1 and the fact that scaling by $\sqrt{1 - \sigma^2}$ and convolving by $\mathcal{N}(0, \sigma^2)$ incurs $O(\sigma) = o(1/\sqrt{m})$ in Wasserstein, we conclude that $W_1(P_v^{D'}, P_v^A) = W_1(D', A) = o(1/\sqrt{m})$ and similarly for $W_1(P_{v'}^{D'}, P_{v'}^A)$. So by triangle inequality for Wasserstein, $W_1(P_v^{D'}, P_{v'}^{D'}) = \Omega(1/\sqrt{m})$ as claimed. $\qquad\square$

We conclude that in Theorem 3.2, the distribution $D$ also satisfies the Wasserstein analogue of Part 3, i.e. $W_1(P_v^D, P_{v'}^D) \ge \Omega(1/\sqrt{m})$ for any $v, v' \in \mathbb{S}^{d-1}$ satisfying $|\langle v, v' \rangle| \ge 1/2$. We can now prove an analogue of Theorem 3.1 for Wasserstein:

**Theorem C.3.** *Let $d \in \mathbb{N}$ be sufficiently large. Any SQ algorithm which, given SQ access to an arbitrary one-hidden-layer ReLU network pushforward of $\mathcal{N}(0, \mathrm{Id}_d)$ of size $O(\log d / \log \log d)$ with $\mathrm{poly}(d)$-bounded weights, outputs a distribution which is $O(\sqrt{\log \log d / \log d})$-close in $d_{\mathrm{TV}}(\cdot)$ must make at least $d^{\Omega(\log d / \log \log d)}$ queries to either $\mathrm{STAT}(\tau)$ or $\mathrm{VSTAT}(1/\tau^2)$ for $\tau = d^{-\Omega(\log d / \log \log d)}$.*

*Proof.* By Theorem 3.2 applied with sufficiently large odd $m$ and sufficiently small $\sigma$, together with the above consequence of Lemma C.2, there exists a distribution $D = f^*(\mathcal{N}(0, \mathrm{Id}_2))$ over $\mathbb{R}$ for

$f^* : \mathbb{R}^2 \to \mathbb{R}$ of size $O(m)$ with $m^{O(m)}$-bounded weights satisfying the hypotheses of Lemma 3.3 for $\epsilon = O(1/\sqrt{m})$, and $\chi^2(D, \mathcal{N}(0,1)) \leq \exp(O(m))$ (note that while Lemma 3.3 is stated for $\mathrm{d}_{\mathrm{TV}}(\cdot)$, it is also true with $\mathrm{d}_{\mathrm{TV}}(\cdot)$ replaced with Wasserstein-1). As long as $m \leq d^{O(C)}$, we conclude that an SQ algorithm for learning any distribution from $\{P_v^D\}_{v \in \mathbb{S}^{d-1}}$ to Wasserstein-1 distance $O(1/\sqrt{m})$ must make at least $d^{m+1}$ queries to $\mathrm{STAT}(\tau)$ or $\mathrm{VSTAT}(1/\tau^2)$ for $\tau \triangleq O(d)^{-(m+1)(1/4-C/2)} \cdot \exp(O(m))$. By taking $m = \Theta(\log d / \log \log d)$, we ensure that $m^{O(m)} \leq \mathrm{poly}(d)$. We're done by taking $C$ in Lemma 3.3 to be $C = 1/4$. $\qquad\square$

# D   Hardness From Supervised Learning

In this section we make rigorous the claim from the introduction that lower bounds for PAC learning neural networks from Gaussian labeled examples imply lower bounds for learning neural network pushforwards. Formally, consider the following distinguishing problem:

**Definition 8** (Distinguishing labeled examples from Gaussian). *For $d \in \mathbb{N}$, let $\mathcal{C}_d$ be some class of functions from $\mathbb{R}^d$ to $\mathbb{R}$. The learner is given $\mathrm{poly}(d)$ many samples $(x_1, y_1), \ldots, (x_N, y_N)$ where $x_1, \ldots, x_N$ are independent draws from $\mathcal{N}(0, \mathrm{Id}_d)$ such that one of the following is true: 1) there is some $h \in \mathcal{C}$ for which $y_i = h(x_i)$ for all $i \in [N]$, or 2) every $y_i$ is an independent sample from $\mathcal{N}(0, 1)$. We say that an algorithm distinguishes between these two situations with constant advantage if the probability it outputs YES (resp. NO) under the former (resp. latter) is at least $2/3$, where the probability is with respect to the randomness of the samples and internal randomness of the algorithm.*

Here we make the simple observation that an oracle for distinguishing any given family of non-Gaussian pushforwards from $\mathcal{N}(0, \mathrm{Id})$ (an easier task than actually learning pushforwards) immediately implies an algorithm for the distinguishing task in Definition 8.

**Lemma D.1.** *For $d \in \mathbb{N}$, let $\mathcal{C}_d$ be any function class from $\mathbb{R}^d \to \mathbb{R}$ for which the indexing functions $f^{[j]}$, given by $f^{[j]}(x) = x_j$ for some $j \in [d]$, are elements of $\mathcal{C}$. Suppose that for any $d_1, d_2 = \mathrm{poly}(d)$, there is a $\mathrm{poly}(d)$-time algorithm $\mathcal{A}$ for the following task. Let $d_1, d_2 = \mathrm{poly}(d)$, and let $\mathcal{S}$ be a known set of functions $f : \mathbb{R}^{d_1} \to \mathbb{R}^{d_2}$ whose output coordinates are all elements of $\mathcal{C}_{d_1}$ and such that for any $f \in \mathcal{S}$, $\mathrm{d}_{\mathrm{TV}}(f(\mathcal{N}(0, \mathrm{Id}_{d_1})), \mathcal{N}(0, \mathrm{Id}_{d_2})) \geq 1/2$. Then $\mathcal{A}$ can distinguish with constant advantage whether it is given $\mathrm{poly}(d)$ samples from $f(\mathcal{N}(0, \mathrm{Id}_{d_1}))$ for some $f \in \mathcal{S}$ versus samples from $\mathcal{N}(0, \mathrm{Id}_{d_2})$.*

*Under this hypothesis, there is a $\mathrm{poly}(d)$-time algorithm that solves the distinguishing problem of Definition 8 to constant advantage.*

*Proof.* Note that in situation 1) of Definition 8, the joint distribution over $(x, y)$ is given by the pushforward $f(\mathcal{N}(0, \mathrm{Id}))$ where $f : \mathbb{R}^{d+1} \to \mathbb{R}^{d+1}$ is as follows: the first $d$ output coordinates are given by the $d$ indexing functions $f^{[1]}, \ldots, f^{[d]}$, and the last output coordinate is given by $h$. By taking $\mathcal{S}$ in the hypothesis to consist of such $f$, we can thus apply the algorithm $\mathcal{A}$ to distinguish between the two situations in Definition 8 to constant advantage. $\qquad\square$

Note that the contrapositive of the above lemma implies that any lower bound for the task in Definition 8 immediately implies a lower bound for learning pushforwards. While the aforementioned lower bounds of [CGKM22, DV21], which apply when $\mathcal{C}_d$ is the family of neural networks with at least two hidden layers and polynomially bounded size and weights, do not show hardness for the task in Definition 8, note that hardness for this task immediately implies hardness for PAC learning $\mathcal{C}_d$ from Gaussian examples. Indeed, given an algorithm $\mathcal{A}$ that, given $(x_1, h(x_1)), \ldots, (x_N, h(x_N))$, outputs a predictor $\widehat{h}$ for which $\mathbb{E}_g[(h(g) - \widehat{h}(g))^2]$ is small, one can easily solve the task in Definition 8 by running $\mathcal{A}$ and estimating the square loss of the predictor from some fresh samples. In situation 2) of Definition 8, because the labels are random, no predictor can achieve low square loss. So the algorithm which outputs YES if and only if the empirical square loss on fresh samples is small will distinguish between the two situations with constant advantage.

In other words, showing hardness of Definition 8 for $\mathcal{C}_d$ would be a *stronger result* than what is already shown in [CGKM22, DV21]. Putting this and Lemma D.1 together, we conclude that even this stronger hardness result would only imply hardness for learning pushforwards given by $f$ whose output coordinates are functions in $\mathcal{C}_d$ given by neural networks with at least two hidden layers and

polynomially bounded size and weights. In contrast, in the present work, we show hardness for *one* hidden layer, *logarithmic* size, and polynomially bounded weights.