# OpenReview forum: "Learning (Very) Simple Generative Models Is Hard"
_NeurIPS.cc/2022/Conference — NeurIPS 2022 Accept_

### Official Review · Reviewer_EJis · 2022-06-28

**Rating:** 6
**Confidence:** 2
**Soundness:** 3 good
**Presentation:** 3 good
**Contribution:** 3 good

**Summary:**

This paper shows that under the statistical query (SQ) model, no polynomial-time algorithm can solve the unsupervised learning problem even when the output coordinates of F are one-hidden-layer ReLU networks with log(d) neurons.

**Questions:**

Questions:
1. I would be interested to see a discussion on whether the lower bounds in this paper appear in real world examples. For instance, is it possible that the lower bounds in this paper occurs under conventional optimization with low probability? And what does the lower bound results in this paper have to say about the empirical success of deep generative models?

**Limitations:**

The authors acknowledge their limitations and proposes adequate scope for future work. There is no negative social impact that I am aware of.

**Strengths And Weaknesses:**

Strength:
1. Previously, the best lower bounds for this problem simply followed from lower bounds for supervised learning and required at least two hidden layers and poly(d) neurons.

Weakness:
1. This paper should have more discussion about the significance of the lower bounds. Please see my questions below for more about this aspect.

---

> ### Author Response · Authors · 2022-08-02
> **Thanks for raising an important question!**
>
> We thank the reviewer for raising an important question regarding how to interpret our lower bounds. As we state in our conclusions section, “our lower bound construction is worst case, and so it is unlikely to actually arise in practice.” Given the striking dearth of provable guarantees in this literature, it is a priori unclear even what theoretical assumptions are needed to understand why pushforwards are efficiently learnable, let alone explain the empirical successes of deep generative modeling. We thus view our work as an important first step towards putting these kinds of questions on rigorous footing. Analogous results on learning Gaussian mixtures, e.g. the work of [Diakonikolas-Kane-Stewart '17], have been very influential in recent work on the computational complexity of provably learning mixture models, and we hope our contribution can similarly motivative subsequent work on the timely question of provably learning generative models.

---

### Official Review · Reviewer_xmvR · 2022-07-08

**Rating:** 7
**Confidence:** 3
**Soundness:** 4 excellent
**Presentation:** 3 good
**Contribution:** 3 good

**Summary:**

The authors study the family of distributions defined by pushing forward a standard Gaussian through a one-hidden-layer ReLU network with $\mathcal{O}(\log d / \log \log d)$ hidden units and $\mathrm{poly}(d)$ bounded weights. They prove that under the statistical query (SQ) model, there exists a subfamily of such distributions that are arbitrarily pathological, in the sense that learning any approximating distribution is computationally expensive, namely requiring $d^{\omega(1)}$ time and samples.

The authors achieve this by reducing the problem to distinguishing a standard $d$-dimensional Gaussian from a distribution that is a product of a 1D distribution whose first $k$ moments match the moments of a Gaussian along a linear subspace of $\mathbb{R}^d$ and a standard $d-1$-dimensional Gaussian over the orthogonal complement. [1] previously showed this problem to be hard under the SQ model; thus, the main technical contribution of the authors is to show that the latter kind of distribution can be constructed as the pushforward of a small ReLU network.

[1] Ilias Diakonikolas, Daniel M Kane, and Alistair Stewart. Statistical query lower bounds for robust estimation of high-dimensional Gaussians and Gaussian mixtures. In IEEE FOCS 2017.

**Questions:**

 - L201: Why can we absorb $\sigma$ into the $\exp(\mathcal{O}(m))$ term?
 - Do the authors have an intuition regarding what assumptions are "missing" / what is "too loose" that allows the pathological behaviour? My initial guess is that it would be that $\mathrm{poly}(d)$ bounded weights are still too flexible, allowing the models to "overfit" to the moment-based construction. As I understand, in practice, the weight distribution of models trained using MAP inference is usually heavily concentrated around 0.
 - As a follow-up: do the authors expect any part of their bound to be tight (within the "class" of worst-case bounds)? Why / why not?

**Limitations:**



**Strengths And Weaknesses:**

# Strengths
The authors study a very relevant problem: the computational hardness of learning an arbitrary distribution using simple generative models. Their results show that this is a hard problem without making further assumptions, even if the generative model is relatively simple. In particular, the bounds derived are much tighter than previous bounds across every parameter considered, and I believe it is a solid contribution to the literature.

The ODE-based construction to attain the desired mixture of bump functions used to construct the pathological distribution is elegant. However, I cannot judge how transferable this trick is to related problems, as I am not familiar enough with the literature.

I carefully checked the proofs in the main text and skimmed the ones in the supplementary. Beyond what I believe are typos, I am confident that the results are correct.

# Weaknesses
The paper is well-written, and I do not believe it has any significant flaws.

Typos:
 - L51: "networks" -> "network"
 - L132: "...which are known to be computationally hard to known both..."
 - L202: Since the TVD is getting bounded, I believe the upper bound should be $\epsilon$, not $C$, so the line should have $0.49$ instead of $1/4$.
 - L280 and L293: The order of the elements in the triplets is inconsistent.
 - Eq (2) and Eq below L301: The second lower indices should be $\ell$s, not $k$s.
 - L362: first Eq should be $\sqrt{1 - \sigma^2}f(z_1) + \sigma z_2$

---

> ### Author Response · Authors · 2022-08-02
> **Thanks for the positive feedback!**
>
> We thank the reviewer for their positive feedback and careful reading of our paper. In the final version of the paper we will fix the typos pointed out. To address the three questions:
> - The reason we can absorb the $\sigma$ is that we can take $\sigma$ to be $\exp(-\Theta(m))$, and we only pay an exponentially small amount for that in the TV between any $P^D_v$ and $P^D_{v’}$ from bullet point 3 of Theorem 3.2. We will clarify this point in the final version.
> - The suggestion of having a more stringent bound on the weights is a very interesting one! However, note that if we simply scaled all the weights down by a factor of $\textrm{poly}(d)$ to ensure that weights are $O(1)$, the resulting family of pushforwards would still be hard to distinguish from $\mathcal{N}(0,1/\textrm{poly}(d))$, even though the Wasserstein distance would merely contract from the claimed bound in Lemma C.1 by a polynomial factor. As for what assumptions are “missing”, rather than posit a constraint like a bound on the weights or size of the network, we suspect that (as we allude to briefly in the conclusions section) any reasonable smoothed analysis model should suffice to circumvent our lower bound.
> - We conjecture that for networks where every output coordinate is a one-hidden-layer network with $s$ neurons, the lower bound should be $d^{\Omega(s)}$, as matching $s$ moments amounts to $s$ constraints for which we need $s$ degrees of freedom. Our lower bound would thus be tight in the regime where $s$ is logarithmic.

---

> > ### Comment · Reviewer_xmvR · 2022-08-04
> > **Response to the authors**
> >
> > I thank the authors for their rebuttal, especially the intuition they give for my questions.
> >
> > After reading their rebuttal as well as the other reviews, I maintain my opinion that this work will be a solid contribution to the literature and I keep my score.

---

### Official Review · Reviewer_wYoe · 2022-07-09

**Rating:** 8
**Confidence:** 3
**Soundness:** 4 excellent
**Presentation:** 3 good
**Contribution:** 3 good

**Summary:**

The paper is concerned with the computational complexity of learning distributions $f(\mathcal{N}(0, \mathrm{Id}))$ in TV distance where $f$ is an unknown feed forward neural network with ReLU activations. What we already know about this problem: 1) When the network has 0 hidden layers, this can be efficiently solved; 2) Previous lower bounds show SQ lower bounds for 2 hidden layers and polynomially many hidden units. This paper improves on these lower bounds to show a SQ lower bound with 1 hidden layer and logarithmic number of hidden units.

Their proof technique follows now standard ideas introduced in DKS17. To show this SQ lower bound, it sufficies to show that there exists feed forward networks such that the pushforward distribution $f(\mathcal{N}(0, \mathrm{Id}))$ is an arbitrary distribution D in a direction v and standard Gaussian in all directions orthogonal to v. This is enough because DKS17 showed that if D’s moments match those of $\mathcal{N}(0, 1)$ up to some degree $m$, then any SQ algorithm for this task requires at least $d^{m}$ queries.

The goal therefore is thus to construct a network whose pushforward matches the low-degree moments of $\mathcal{N}(0, 1)$. Note that if we were not concerned with the norm of the weights used in the construction, this essentially follows from DKS17. They showed that there exists $m$ weights $\lambda_i$ and $m$ points $h_i$ such that $\sum \lambda_i h_i$ matches the first $2m$ moments of $\mathcal{N}(0, 1)$. We can partition the real line into intervals $I_i$ such that measure of $I_i$ is exactly the weight $\lambda_i$. And therefore, the step function $f(z) = \sum h_i . \mathbb{1}[z \in I_i]$ matches the first $2m$ moments of $\mathcal{N}(0, 1)$, which can be simulated by feedforward network with unbounded weights.

Therefore, the main challenge here is to simulating this distribution approximately with bounded weights. There are two main ideas for achieving this. Firstly,  instead of the step function, which do not have any gaps between the different jumps, we need to construct functions with gaps (of order $m^{-3/2}$) between their jumps (called bumps) so that the slope of these edges can be made small. This is illustrated in Figure 1 and proven in Lemma 4.4 (inspired by DKS17).

Secondly, they need to show that the gap above is enough to decrease the slope. This is achieved by considering an ODE relating the moments $M(t)$ to the height $h(t)$ and slant parameter $\epsilon(t)$. To ensure the moments do not change, they evolve $h(t)$ and $\epsilon(t)$ in the direction with $0$ gradient (which keeps the moment constant). Both existence and bounded $h(t)$ follows from analysis the solutions of the ODE (and showing that certain matrix in the analysis is well conditioned).

**Questions:**

Other than the small typos above, I do not have any other questions.

**Limitations:**

The authors have adequately addressed the limitations and potential negative societal impact of their work.

**Strengths And Weaknesses:**

The paper is written very nicely. It was easy to follow even though there is a lot of technical detail. I thank the authors for the effort they spent on this. Few (subjective) comments on the writing however:
1. Line 76: "Let D be a known, non-Gaussian distribution D over R." This does not read well.
2. Lemma 4.1 does not mention sum lambda =1 but Corollary 4.2 uses this.
3. Before 298-308, it will be helpful to briefly discuss the goal of this analysis. Maybe something like this: "we would like to choose h(t), eps(t) such that the the low-degree moments of ft defined in (1) are constant in t as desired. We now discuss how to choose such h(t) and eps(t)."
4. It will be useful to clarify in the terminology section that $A^{-\top} = (A^{\top})^{-1}$.

Understanding the computational complexity of learning distributions is an important problem. Without any assumptions, most of the learning problems are hard and therefore the constant struggle is to design assumptions which allow us to evade the hard instances and identify the easy instances. The hardness result from this work shows that we might need more assumptions (like smoothed analysis) for efficient learning.

---

> ### Author Response · Authors · 2022-08-02
> **Thank you for the positive review and close reading!**
>
> We thank the reviewer for their very positive review and careful reading of our paper. We are encouraged that they agree that understanding the computational complexity of distribution learning is an important direction, and we will incorporate the writing suggestions in the final version of the paper.

---

### Official Review · Reviewer_nCQE · 2022-07-12

**Rating:** 7
**Confidence:** 3
**Soundness:** 4 excellent
**Presentation:** 2 fair
**Contribution:** 3 good

**Summary:**

The authors consider the problem of learning a one hidden-layer generative model. A one hidden-layer generative model is a neural network with a hidden layer, ReLU activations, then an output layer that takes as input a sample of a standard Gaussian. The goal is to typically learn the parameters, though it also may be to describe the distribution in other ways.

The authors consider the statistical query (SQ) framework. Here an algorithm is allowed to make "statistical queries," which are specified by a function f and return a good estimate of the value E_x[f(x)]. It is often possible to establish unconditional lower bounds in the SQ framework. The SQ framework doesn't directly imply hardness in other senses, though many standard learning algorithms can be adapted to the SQ framework. Thus hardness in this framework at the very least implies that a large class of algorithms can't be used for this problem.

The authors are able to establish that the problem of learning a one hidden-layer generative model is hard in the SQ framework. They start by using a known lower bound by [DKS17] that shows hardness of learning an n-dimensional distribution that is (1) a product distribution in the right basis and (2) the first coordinate in this basis has many moments that match a standard Gaussian and the remaining coordinates are standard Gaussians. The authors are able to construct a family of one hidden-layer generative models that match this description, which implied hardness of learning in the SQ framework.

The authors approach to creating the network is by perturbing an initial solution that may have a description size that is too large. This is described by a differential equation, which they show has a well defined solution.

**Questions:**

Are there any known information-theoretic bounds that show that there exist algorithms that use polynomial samples (but potentially exponential time) for solving this problem?

How important is the use of ReLU activations for the result? Is it straightforward to generalize to other activations?



**Limitations:**

The authors mentioned that hardness in the SQ framework does not exactly capture other notions of hardness and only (1) shows that a class of algorithms would not work for the problem and (2) only suggests hardness in other ways.

Additionally, the authors mention that their results only apply to worst case problems and leave open practical cases where learning is possible.

**Strengths And Weaknesses:**

Strengths:

(1) The paper solves an interesting problem. Learning generative models with just an output layer followed by ReLU activations is known to be solvable. Generative models with two hidden layers are known to be hard (though under different assumptions than this work). This work explores that natural missing case of networks with one hidden layer.

(2) The paper method of designing the hard class of networks is quite rigorous and advanced and can potentially be useful in future work.

Weaknessess:

(1) The biggest weakness is the presentation. The intro is well written, however the subsequent sections can be quite dense and often become a list of lemmas. While some of the complexity is necessary, I believe the authors could do a better job to share their intuition about the result. Reading related work often describes the necessary background with more clarity.

One related work not cited is:
Vempala, Santosh, and John Wilmes. "Gradient descent for one-hidden-layer neural networks: Polynomial convergence and SQ lower bounds." Conference on Learning Theory. PMLR, 2019.

---

> ### Author Response · Authors · 2022-08-02
> **Thanks for the positive feedback!**
>
> We thank the reader for their positive feedback and for engaging closely with our work. We are encouraged that they find the problem and techniques interesting. We will make sure to use the extra page allotted for the camera ready to provide ample breathing room for the technical sections and to set up the necessary background more clearly. We will also add some discussion about [Vempala-Wilmes ‘19]; note that that paper studies a somewhat different setting, namely supervised learning of polynomials over the uniform distribution on the unit sphere. Furthermore, they only give CSQ lower bounds, and to obtain superpoly lower bounds, their functions require superpoly description length.
>
> Here we respond to the two questions raised:
> - To obtain an information-theoretic bound, one could simply grid over parameters and apply Scheffe selection. For pushforwards given by a function $F: \mathbb{R}^d\to\mathbb{R}^{d’}$ where every one of the output coordinates is a one-hidden layer network with $s$ neurons and bounded weights, there are $dd’s$ parameters in total, so one can construct an $\epsilon$-net over such networks with size exponential in $dd’s$. The sample complexity for Scheffe selection will then be logarithmic in the size of this net and thus polynomial in all parameters.
> - Our arguments should extend without too much effort to reasonable piecewise linear activations. It is an intriguing open question whether one can extend our results to other activations like sigmoid or polynomial, but as our starting point is to implement the univariate moment-matching construction via step functions, one would need rather different arguments.

---

> > ### Comment · Reviewer_nCQE · 2022-08-09
> > **Thank you for your response**
> >
> > Thank you for your response and new information-theoretic bound. I think it complements the results presented well.

---

### Meta-Review · Area_Chair_7d2Y · 2022-09-03

**Recommendation:** Accept
**Confidence:** Certain

**Metareview:**

The paper present an SQ lower bound for the problem of learning one-hidden layer ReLU network generative models of logarithmic size.

**Award:**

No

---

### Decision · Program_Chairs · 2022-09-14

Accept